# Outlier Robust Mean Estimation with Subgaussian Rates via Stability

**Ilias Diakonikolas**
University of Wisconsin-Madison
ilias@cs.wisc.edu

**Daniel M. Kane**
University of California, San Diego
dakane@cs.ucsd.edu

**Ankit Pensia**
University of Wisconsin-Madison
ankitp@cs.wisc.edu

## Abstract

We study the problem of outlier robust high-dimensional mean estimation under a finite covariance assumption, and more broadly under finite low-degree moment assumptions. We consider a standard stability condition from the recent robust statistics literature and prove that, except with exponentially small failure probability, there exists a large fraction of the inliers satisfying this condition. As a corollary, it follows that a number of recently developed algorithms for robust mean estimation, including iterative filtering and non-convex gradient descent, give optimal error estimators with (near-)subgaussian rates. Previous analyses of these algorithms gave significantly suboptimal rates. As a corollary of our approach, we obtain the first computationally efficient algorithm with subgaussian rate for outlier-robust mean estimation in the strong contamination model under a finite covariance assumption.

## 1 Introduction

### 1.1 Background and Motivation

Consider the following problem: For a given family $\mathcal{F}$ of distributions on $\mathbb{R}^d$, estimate the mean of an unknown $D \in \mathcal{F}$, given access to i.i.d. samples from $D$. This is the problem of (multivariate) mean estimation and is arguably *the* most fundamental statistical task. In the most basic setting where $\mathcal{F}$ is the family of high-dimensional Gaussians, the empirical mean is well-known to be an optimal estimator — in the sense that it achieves the best possible accuracy-confidence tradeoff and is easy to compute. Unfortunately, the empirical mean is known to be highly suboptimal if we relax the aforementioned modeling assumptions. In this work, we study high-dimensional mean estimation *in the high confidence regime* when the underlying family $\mathcal{F}$ is only assumed to satisfy bounded moment conditions (e.g., finite covariance). Moreover, we relax the "i.i.d. assumption" and aim to obtain estimators that are robust to a constant fraction of adversarial outliers.

Throughout this paper, we focus on the following data contamination model (see, e.g., [DKK+16]) that generalizes several existing models, including Huber's contamination model [Hub64].

**Definition 1.1** (Strong Contamination Model). *Given a parameter $0 < \epsilon < 1/2$ and a distribution family $\mathcal{F}$ on $\mathbb{R}^d$, the* adversary *operates as follows: The algorithm specifies the number of samples $n$, and $n$ samples are drawn from some unknown $D \in \mathcal{F}$. The adversary is allowed to inspect the samples, remove up to $\epsilon n$ of them and replace them with arbitrary points. This modified set of $n$ points is then given as input to the algorithm. We say that a set of samples is $\epsilon$-corrupted if it is generated by the above process.*

The parameter $\epsilon$ in Definition 1.1 is the fraction of outliers and quantifies the power of the adversary. Intuitively, among our input samples, an unknown $(1 - \epsilon)$ fraction are generated from a distribution of interest and are called *inliers*, and the rest are called *outliers*.

We note that the strong contamination model is strictly stronger than Huber's contamination model. Recall that in Huber's contamination model [Hub64], the adversary generates samples from a mixture distribution $P$ of the form $P = (1 - \epsilon)D + \epsilon N$, where $D \in \mathcal{F}$ is the unknown target distribution and $N$ is an adversarially chosen noise distribution. That is, in Huber's model the adversary is oblivious to the inliers and is only allowed to add outliers.

In the context of robust mean estimation, we want to design an algorithm (estimator) with the following performance: Given any $\epsilon$-corrupted set of $n$ samples from an unknown distribution $D \in \mathcal{F}$, the algorithm outputs an estimate $\widehat{\mu} \in \mathbb{R}^d$ of the target mean $\mu$ of $D$ such that *with high probability* the $\ell_2$-norm $\|\widehat{\mu} - \mu\|$ is small. The ultimate goal is to obtain a *computationally efficient estimator with optimal confidence-accuracy tradeoff*. For concreteness, in the proceeding discussion we focus on the case that $\mathcal{F}$ is the family of all distributions on $\mathbb{R}^d$ with bounded covariance, i.e., any $D \in \mathcal{F}$ has covariance matrix $\Sigma \preceq I$. (We note that the results of this paper apply for the more general setting where $\Sigma \preceq \sigma^2 I$, where $\sigma > 0$ is unknown to the algorithm.)

Perhaps surprisingly, even for the special case of $\epsilon = 0$ (i.e., without adversarial contamination), designing an optimal mean estimator in the high-confidence regime is far from trivial. In particular, it is well-known (and easy to see) that the empirical mean achieves highly sub-optimal rate. A sequence of works in mathematical statistics (see, e.g., [Cat12, Min15, DLLO16, LM19c]) designed novel estimators with improved rates, culminating in an optimal estimator [LM19c]. See [LM19a] for a survey on the topic. The estimator of [LM19c] is based on the median-of-means framework and achieves a "subgaussian" performance guarantee:

$$\|\widehat{\mu} - \mu\| = O(\sqrt{d/n} + \sqrt{\log(1/\tau)/n}) \, , \tag{1}$$

where $\tau > 0$ is the failure probability. The error rate (1) is information-theoretically optimal for any estimator and matches the error rate achieved by the empirical mean on Gaussian data. Unfortunately, the estimator of [LM19c] is not efficiently computable. In particular, known algorithms to compute it have running time exponential in the dimension $d$. Related works [Min15, PBR19] provide computationally efficient estimators alas with suboptimal rates. The first polynomial time algorithm achieving the optimal rate (1) was given in [Hop20], using a convex program derived from the Sums-of-Squares method. Efficient algorithms with improved asymptotic runtimes were subsequently given in [CFB19, DL19, LLVZ20].

We now turn to the outlier-robust setting ($\epsilon > 0$) for the constant confidence regime, i.e., when the failure probability $\tau$ is a small universal constant. The statistical foundations of outlier-robust estimation were laid out in early work by the robust statistics community, starting with the pioneering works of [Tuk60] and [Hub64]. For example, the minimax optimal estimator satisfies:

$$\|\widehat{\mu} - \mu\| = O(\sqrt{\epsilon} + \sqrt{d/n}) \, . \tag{2}$$

Until fairly recently however, all known polynomial-time estimators attained sub-optimal rates. Specifically, even in the limit when $n \to \infty$, known polynomial time estimators achieved error of $O(\sqrt{\epsilon d})$, i.e., scaling polynomially with the dimension $d$. Recent work in computer science, starting with [DKK$^+$16, LRV16], gave the first efficiently computable outlier-robust estimators for high-dimensional mean estimation. For bounded covariance distributions, [DKK$^+$17, SCV18] gave efficient algorithms with the right error guarantee of $O(\sqrt{\epsilon})$. Specifically, the filtering algorithm of [DKK$^+$17] is known to achieve a near-optimal rate of $O(\sqrt{\epsilon} + \sqrt{d \log d/n})$.

In this paper, we aim to achieve the best of both worlds. In particular, we ask the following question:

*Can we design computationally efficient estimators with subgaussian rates and optimal dependence on the contamination parameter $\epsilon$?*

Recent work [LM19b] gave an *exponential time* estimator with optimal rate in this setting. Specifically, [LM19b] showed that a multivariate extension of the trimmed-mean achieved the optimal error of

$$\|\widehat{\mu} - \mu\| = O(\sqrt{\epsilon} + \sqrt{d/n} + \sqrt{\log(1/\tau)/n}) \, . \tag{3}$$

We note that [LM19b] posed as an open question the existence of a computationally efficient estimator achieving the optimal rate (3). Two recent works [DL19, LLVZ20] gave efficient estimators with

subgaussian rates that are outlier-robust in the *additive* contamination model — a *weaker* model than that of Definition 1.1. Prior to this work, no polynomial time algorithm with optimal (or near-optimal) rate was known in the strong contamination model of Definition 1.1. As a corollary of our approach, we answer the question of [LM19b] in the affirmative (see Proposition 1.6). In the following subsection, we describe our results in detail.

## 1.2 Our Contributions

At a high-level, the main conceptual contribution of this work is in showing that several previously developed computationally efficient algorithms for high-dimensional robust mean estimation achieve near-subgaussian rates or subgaussian rates (after a simple pre-processing). A number of these algorithms are known to succeed under a standard *stability* condition (Definition 1.2) – a simple deterministic condition on the empirical mean and covariance of a finite point set. We will call such algorithms *stability-based*.

Our contributions are as follows:

- We show (Theorem 1.4) that given a set of i.i.d. samples from a finite covariance distribution, except with exponentially small failure probability, there exists a large fraction of the samples satisfying the stability condition. As a corollary, it follows (Proposition 1.5) that *any* stability-based robust mean estimation algorithm achieves optimal error with (near-)subgaussian rates.

- We show an analogous probabilistic result (Theorem 1.8) for known covariance distributions (or, more generally, spherical covariance distributions) with bounded $k$-th moment, for some $k \geq 4$. As a corollary, we obtain that *any* stability-based robust mean estimator achieves optimal error with (near-)subgaussian rates (Proposition 1.9.)

- For the case of finite covariance distributions, we show (Proposition 1.6) that a simple pre-processing step followed by any stability-based robust mean estimation algorithm yields optimal error and subgaussian rates.

To formally state our results, we require some terminology and background.

**Basic Notation**    For a vector $v \in \mathbb{R}^d$, we use $\|v\|$ to denote its $\ell_2$-norm. For a square matrix $M$, we use $\mathrm{tr}(M)$ to denotes its trace, and $\|M\|$ to denote its spectral norm. We say a symmetric matrix $A$ is PSD (positive semidefinite) if $x^T A x \geq 0$ for all vectors $x$. For a PSD matrix $M$, we use $\mathrm{r}(M)$ to denote its stable rank (or intrinsic dimension), i.e., $\mathrm{r}(M) := \mathrm{tr}(M)/\|M\|$. For two symmetric matrices $A$ and $B$, we use $\langle A, B \rangle$ to denote the trace inner product $\mathrm{tr}(AB)$ and say $A \preceq B$ when $B - A$ is PSD.

We use $[n]$ to denote the set $\{1, \ldots, n\}$ and $\mathcal{S}^{d-1}$ to denote the $d$-dimensional unit sphere. We use $\Delta_n$ to denote the probability simplex on $[n]$, i.e., $\Delta_n = \{w \in \mathbb{R}^n : w_i \geq 0, \sum_{i=1}^n w_i = 1\}$. For a multiset $S = \{x_1, \ldots, x_n\} \subset \mathbb{R}^d$ of cardinality $n$ and $w \in \Delta_n$, we use $\mu_w$ to denote its weighted mean $\mu_w = \sum_{i=1}^n w_i x_i$. Similarly, we use $\overline{\Sigma}_w$ to denote its weighted second moment matrix (centered with respect to $\mu$) $\overline{\Sigma}_w = \sum_{i=1}^n w_i (x_i - \mu)(x_i - \mu)^T$. For a set $S \subset \mathbb{R}^d$, we denote $\mu_S = (1/|S|) \sum_{x \in S} x$ and $\overline{\Sigma}_S = (1/|S|) \sum_{x \in S} (x - \mu)(x - \mu)^T$ to denote the mean and (central) second moment matrix with respect to the uniform distribution on $S$.

**Stability Condition and Robust Mean Estimation.**    We can now define the stability condition:
**Definition 1.2** (see, e.g., [DK19]). *Fix $0 < \epsilon < 1/2$ and $\delta \geq \epsilon$. A finite multiset $S \subset \mathbb{R}^d$ is $(\epsilon, \delta)$-stable with respect to mean $\mu \in \mathbb{R}^d$ and $\sigma^2$ if for every $S' \subseteq S$ with $|S'| \geq (1 - \epsilon)|S|$, the following conditions hold: (i) $\|\mu_{S'} - \mu\| \leq \sigma \delta$, and (ii) $\|\overline{\Sigma}_{S'} - \sigma^2 I\| \leq \sigma^2 \delta^2/\epsilon$.*

The aforementioned condition or a variant thereof is used in every known outlier-robust mean estimation algorithm. Definition 1.2 requires that after restricting to a $(1 - \epsilon)$-density subset $S'$, the sample mean of $S'$ is within $\sigma\delta$ of the mean $\mu$, and the sample variance of $S'$ is $\sigma^2(1 \pm \delta^2/\epsilon)$ in every direction. (We note that Definition 1.2 is intended for distributions with covariance $\Sigma \preceq \sigma^2 I$). We will omit the parameters $\mu$ and $\sigma^2$ when they are clear from context. In particular, our proofs will focus on the case $\sigma^2 = 1$, which can be achieved by scaling the datapoints appropriately.

A number of known algorithmic techniques previously used for robust mean estimation, including convex programming based methods [DKK$^+$16, SCV18, CDG19], iterative filtering [DKK$^+$16,

DKK⁺17, DHL19], and even first-order methods [CDGS20, ZJS20], are known to succeed under the stability condition. Specifically, prior work has established the following theorem:

**Theorem 1.3** (Robust Mean Estimation Under Stability, see, e.g., [DK19]). *Let $T \subset \mathbb{R}^d$ be an $\epsilon$-corrupted version of a set $S$ with the following properties: $S$ contains a subset $S' \subseteq S$ such that $|S'| \geq (1 - \epsilon)|S|$ and $S'$ is $(C\epsilon, \delta)$ stable with respect to $\mu \in \mathbb{R}^d$ and $\sigma^2$, for a sufficiently large constant $C > 0$. Then there is a polynomial-time algorithm, that on input $\epsilon, T$, computes $\widehat{\mu}$ such that $\|\widehat{\mu} - \mu\| = O(\sigma\delta)$.*

We note in particular that the iterative filtering algorithm [DKK⁺17, DK19] (see also Section 2.4.3 of [DK19]) is a very simple and practical stability-based algorithm. While previous works made the assumption that the upper bound parameter $\sigma^2$ is known to the algorithm, we point out in Appendix A.2 that essentially the same algorithm and analysis works for unknown $\sigma^2$ as well.

**Our Results.** Our first main result establishes the stability of a subset of i.i.d. points drawn from a distribution with bounded covariance.

**Theorem 1.4.** *Fix any $0 < \tau < 1$. Let $S$ be a multiset of $n$ i.i.d. samples from a distribution on $\mathbb{R}^d$ with mean $\mu$ and covariance $\Sigma$. Let $\epsilon' = \Theta(\log(1/\tau)/n + \epsilon) \leq c$, for a sufficiently small constant $c > 0$. Then, with probability at least $1 - \tau$, there exists a subset $S' \subseteq S$ such that $|S'| \geq (1 - \epsilon')n$ and $S'$ is $(2\epsilon', \delta)$-stable with respect to $\mu$ and $\|\Sigma\|$, where $\delta = O(\sqrt{(\mathrm{r}(\Sigma) \log \mathrm{r}(\Sigma))/n} + \sqrt{\epsilon} + \sqrt{\log(1/\tau)/n})$.*

Theorem 1.4 significantly improves the probabilistic guarantees in prior work on robust mean estimation. This includes the *resilience* condition of [SCV18, ZJS19] and the *goodness* condition of [DHL19].

As a corollary, it follows that any stability-based algorithm for robust mean estimation achieves near-subgaussian rates.

**Proposition 1.5.** *Let $T$ be an $\epsilon$-corrupted set of $n$ samples from a distribution in $\mathbb{R}^d$ with mean $\mu$ and covariance $\Sigma$. Let $\epsilon' = \Theta(\log(1/\tau)/n + \epsilon) \leq c$ be given, for a constant $c > 0$. Then any stability-based algorithm on input $T$ and $\epsilon'$, efficiently computes $\widehat{\mu}$ such that with probability at least $1 - \tau$, we have $\|\widehat{\mu} - \mu\| = O(\sqrt{(\mathrm{tr}(\Sigma) \log \mathrm{r}(\Sigma))/n} + \sqrt{\|\Sigma\|\epsilon} + \sqrt{\|\Sigma\| \log(1/\tau)/n})$.*

We note that the above error rate is minimax optimal in both $\epsilon$ and $\tau$, and the restriction of $\log(1/\tau)/n = O(1)$ is information-theoretically required [DLLO16]. In particular, the term $\sqrt{\log(1/\tau)/n}$ is *additive* as opposed to multiplicative. The first term is near-optimal, up to the $\sqrt{\log \mathrm{r}(\Sigma)}$ factor, which is at most $\sqrt{\log d}$ (recall that $\mathrm{r}(\Sigma)$ denotes the stable rank of $\Sigma$, i.e., $\mathrm{r}(\Sigma) = \mathrm{tr}(\Sigma)/\|\Sigma\|$). Prior to this work, the existence of a polynomial-time algorithm achieving the above near-subgaussian rate in the strong contamination model was open. Proposition 1.5 shows that any stability-based algorithm suffices for this purpose, and in particular it implies that the iterative filtering algorithm [DK19] achieves this rate *as is*.

Given the above, a natural question is whether stability-based algorithms achieve subgaussian rates *exactly*, i.e., whether they match the optimal bound (3) attained by the computationally inefficient estimator of [LM19b]. While the answer to this question remains open, we show that after a simple pre-processing of the data, stability-based estimators are indeed subgaussian.

The pre-processing step follows the median-of-means principle [NU83, JVV86, AMS99]. Given a multiset of $n$ points $x_1, \ldots, x_n$ in $\mathbb{R}^d$ and $k \in [n]$, we proceed as follows:

1. First randomly bucket the data into $k$ disjoint buckets of equal size (if $k$ does not divide $n$, remove some samples) and compute their empirical means $z_1, \ldots, z_k$.

2. Output an (appropriately defined) multivariate median of $z_1, \ldots, z_k$.

Notably, for the case of $\epsilon = 0$, all known efficient mean estimators with subgaussian rates use the median-of-means framework [Hop20, DL19, CFB19, LLVZ20].

To obtain the desired computationally efficient robust mean estimators with subgaussian rates, we proceed as follows:

1. Given a multiset $S$ of $n$ $\epsilon$-corrupted samples, randomly group the data into $k = \lfloor \epsilon'n \rfloor$ disjoint buckets, where $\epsilon' = \Theta(\log(1/\tau)/n + \epsilon)$, and let $z_1, \ldots, z_k$ be the corresponding empirical means of the buckets.

2. Run any stability-based robust mean estimator on input $\{z_1, \ldots, z_k\}$.

Specifically, we show:

**Proposition 1.6.** *(informal) Consider the same setting as in Proposition 1.5. Let $k = \lfloor \epsilon' n \rfloor$ and $z_1, \ldots, z_k$ be the points after median-of-means pre-processing on the corrupted set $T$. Then any stability-based algorithm, on input $\{z_1, \ldots, z_k\}$, computes $\widehat{\mu}$ such that with probability at least $1 - \tau$, it holds $\|\widehat{\mu} - \mu\| = O(\sqrt{\text{tr}(\Sigma)/n} + \sqrt{\|\Sigma\| \epsilon} + \sqrt{\|\Sigma\| \log(1/\tau)/n})$.*

Proposition 1.6 yields the first computationally efficient algorithm with subgaussian rates in the strong contamination model, answering the open question of [LM19b].

To prove Proposition 1.6, we establish a connection between the median-of-means principle and stability. In particular, we show that the key probabilistic lemma from the median-of-means literature [LM19c, DL19] also implies stability.

**Theorem 1.7.** *(informal) Consider the setting of Theorem 1.4 and set $k = \lfloor \epsilon' n \rfloor$. The set $\{z_1, \ldots, z_k\}$, with probability $1 - \tau$, contains a subset of size at least $0.99k$ which is $(0.1, \delta)$-stable with respect to $\mu$ and $k\|\Sigma\|/n$, where $\delta = O(\sqrt{\text{r}(\Sigma)/k} + 1)$.*

A drawback of the median-of-means framework is that the error dependence on $\epsilon$ does not improve if we impose stronger assumptions on the distribution. Even if the underlying distribution is an identity covariance Gaussian, the error rate would scale as $O(\sqrt{\epsilon})$, whereas the stability-based algorithms achieve error of $O(\epsilon \sqrt{\log(1/\epsilon)})$ [DKK$^+$16]. Our next result establishes tighter error bounds for distributions with identity covariance and bounded central moments.

We say that a distribution has a bounded $k$-th central moment $\sigma_k$, if for all unit vectors $v$, it holds $(\mathbb{E}(v^T(X - \mu))^k)^{1/k} \leq \sigma_k (\mathbb{E}(v^T(X - \mu))^2)^{1/2}$. For such distributions, we establish the following stronger stability condition.

**Theorem 1.8.** *Let $S$ be a multiset of $n$ i.i.d. samples from a distribution on $\mathbb{R}^d$ with mean $\mu$, covariance $\Sigma = I$, and bounded central moment $\sigma_k$, for some $k \geq 4$. Let $\epsilon' = \Theta(\log(1/\tau)/n + \epsilon) \leq c$, for a sufficiently small constant $c > 0$. Then, with probability at least $1 - \tau$, there exists a subset $S' \subseteq S$ such that $|S'| \geq (1 - \epsilon')n$ and $|S'|$ is $(2\epsilon', \delta)$-stable with respect to $\mu$ and $\sigma^2 = 1$, where $\delta = O(\sqrt{d \log d/n} + \sigma_k \epsilon^{1 - \frac{1}{k}} + \sigma_4 \sqrt{\log(1/\tau)/n})$.*

As a corollary, we obtain the following result for robust mean estimation with high probability in the strong contamination model:

**Proposition 1.9.** *Let $T$ be an $\epsilon$-corrupted set of $n$ points from a distribution on $\mathbb{R}^d$ with mean $\mu$, covariance $\sigma^2 I$, and $k$-th bounded central moment $\sigma_k$, for some $k \geq 4$. Let $\epsilon' = \Theta(\log(1/\tau)/n + \epsilon) \leq c$ be given, for some $c > 0$. Then any stability-based algorithm, on input $T$ and $\epsilon'$, efficiently computes $\widehat{\mu}$ such that with probability at least $1 - \tau$, we have $\|\widehat{\mu} - \mu\| = O(\sigma(\sqrt{d \log d/n} + \sigma_k \epsilon^{1 - \frac{1}{k}} + \sigma_4 \sqrt{\log(1/\tau)/n}))$.*

We note that the above error rate is near-optimal up to the $\log d$ factor and the dependence on $\sigma_4$. Prior to this work, no polynomial-time estimator achieving this rate was known. Finally, recent computational hardness results [HL19] suggest that the assumption on the covariance above is inherent to obtain computationally efficient estimators with error rate better than $\Omega(\sqrt{\epsilon})$, even in the constant confidence regime.

## 1.3 Related Work

Since the initials works [DKK$^+$16, LRV16], there has been an explosion of research activity on algorithmic aspects of outlier-robust high dimensional estimation by several communities. See, e.g., [DK19] for a recent survey on the topic. In the context of outlier-robust mean estimation, a number of works [DKK$^+$17, SCV18, CDG19, DHL19] have obtained efficient algorithms under various assumptions on the distribution of the inliers. Notably, efficient high-dimensional outlier-robust mean estimators have been used as primitives for robustly solving machine learning tasks that can be expressed as stochastic optimization problems [PSBR20, DKK$^+$18]. The above works typically focus on the constant probability error regime and do not establish subgaussian rates for their estimators.

Two recent works [DL19, LLVZ20] studied the problem of outlier-robust mean estimation in the additive contamination model (when the adversary is only allowed to add outliers) and gave computationally efficient algorithms with subgaussian rates. Specifically, [DL19] gave an SDP-based algorithm, which is very similar to the algorithm of [CDG19]. The algorithm of [LLVZ20] is a fairly sophisticated iterative spectral algorithm, building on [CFB19]. In the strong contamination model, non-constructive outlier-robust estimators with subgaussian rates were established very recently. Specifically, [LM19b] gave an exponential time estimator achieving the optimal rate. Our Proposition 1.6 implies that a very simple and practical algorithm – pre-processing followed by iterative filtering [DKK+17, DK19] – achieves this guarantee. In an independent and concurrent work, Hopkins, Li, and Zhang [HLZ20] also studied the relation between median-of-means and stability for the case of finite covariance.

**Organization**    In Section 2, we give the proof sketch of Theorem 1.4 that establishes the stability of points sampled from a finite covariance distribution. In Section 3, we briefly comment on the proof sketch of Theorem 1.8, with the detailed proof in Appendix F. We defer the detailed proof of Theorem 1.7 to Appendix B.

## 2    Robust Mean Estimation for Finite Covariance Distributions

**Problem Setting**    Consider a distribution $P$ in $\mathbb{R}^d$ with unknown mean $\mu$ and unknown covariance $\Sigma$. We first note that it suffices to consider the distributions such that $\|\Sigma\| = 1$. Note that for covariance matrices $\Sigma$ with $\|\Sigma\| = 1$, we have $r(\Sigma) = tr(\Sigma)$. In the remainder of this section, we will thus establish the $(\epsilon, \delta)$ stability with respect to $\mu$ and $\sigma^2 = 1$, where $\delta = O(\sqrt{tr(\Sigma)\log(r(\Sigma))/n} + \sqrt{\epsilon} + \sqrt{\log(1/\tau)/n})$.

Let $S$ be a multiset of $n$ i.i.d. samples from $P$. For the ease of exposition, we will assume that the support of $P$ is bounded, i.e., for each $i$, $\|x_i - \mu\| = O(\sqrt{tr(\Sigma)/\epsilon})$ almost surely. A simple reduction, outlined in Appendix E, removes this assumption. We first relax the conditions for stability in the Definition 1.2 in the following Claim 2.1 at an additional cost of $O(\sqrt{\epsilon})$.

**Claim 2.1.** *(Stability for bounded covariance) Let $R \subset \mathbb{R}^d$ be a finite multiset such that $\|\mu_R - \mu\| \leq \delta$, and $\|\overline{\Sigma}_R - I\| \leq \delta^2/\epsilon$ for some $0 \leq \epsilon \leq \delta$. Then $R$ is $(\Theta(\epsilon), \delta')$ stable with respect to $\mu$ (and $\sigma^2 = 1$), where $\delta' = O(\delta + \sqrt{\epsilon})$.*

Given Claim 2.1, our goal in proving Theorem 1.4 is to show that with probability $1 - \tau$, there exists a set $S' \subseteq S$ such that $|S'| \geq (1 - \epsilon')n$, $\|\mu_{S'} - \mu\| \leq \delta$ and $\|\overline{\Sigma}_S - I\| \leq \delta^2/\epsilon'$, for some value of $\delta = O(\sqrt{tr(\Sigma)\log r(\Sigma)/n} + \sqrt{\epsilon} + \sqrt{\log(1/\tau)/n})$ and $\epsilon' = \Theta(\epsilon + \log(1/\tau)/n)$.

We first remark that the original set $S$ of $n$ i.i.d. data points does not satisfy either of the conditions in Claim 2.1. It does not satisfy the first condition because the sample mean is highly sub-optimal for heavy-tailed data [Cat12]. For the second condition, we note that the known concentration results for $\overline{\Sigma}_S$ are not sufficient. For example, consider the case of $\Sigma = I$ in the parameter regime of $\epsilon, \tau$, and $n$ such that $\epsilon = O(\log(1/\tau)/n)$ and $n = \Omega(d \log d/\epsilon)$ so that $\delta = O(\sqrt{\epsilon})$. For $S$ to be $(\epsilon, \delta)$ stable, we require that $\|\overline{\Sigma}_S - I\| = O(1)$ with probability $1 - \tau$. However, the Matrix-Chernoff bound (see, e.g., [Tro15, Theorem 5.1.1]) only guarantees that with probability at least $1 - \tau$, $\|\overline{\Sigma}_S - I\| = \tilde{O}(d)$.

The rest of this section is devoted to showing that, with high probability, it is possible to remove $\epsilon'n$ points from $S$ such that both conditions in Claim 2.1 are satisfied for the subset.

### 2.1    Controlling the Variance

As a first step, we show that it is possible to remove an $\epsilon$-fraction of points so that the second moment matrix concentrates. Since finding a subset is a discrete optimization problem, we first perform a continuous relaxation: instead of finding a large subset, we find a suitable distribution on points. Define the following set of distributions:

$$\Delta_{n,\epsilon} = \left\{ w \in \mathbb{R}^n : 0 \leq w_i \leq 1/((1-\epsilon)n); \sum_{i=1}^{n} w_i = 1 \right\}.$$

Note that $\Delta_{n,\epsilon}$ is the convex hull of all the uniform distributions on $S' \subseteq S : |S'| \geq (1 - \epsilon)n$. In Appendix E.2, we show how to recover a subset $S'$ from the $w$. Although we use the set $\Delta_{n,\epsilon}$

for the sole purpose of theoretical analysis, the object $\Delta_{n,\epsilon}$ has also been useful in the design of computationally efficient algorithms [DKK+16, DK19]. We will now show that, with high probability, there exists a $w \in \Delta_{n,\epsilon}$ such that $\overline{\Sigma}_w$ has small spectral norm.

Our proof technique has three main ingredients: (i) minimax duality, (ii) truncation, and (iii) concentration of truncated empirical processes. Let $\mathcal{M}$ be the set of all PSD matrices with trace norm 1, i.e., $\mathcal{M} = \{M : M \succeq 0, \mathrm{tr}(M) = 1\}$. Using minimax duality [Sio58] and the variational characterization of spectral norm, we obtain the following reformulation:

$$\min_{w \in \Delta_{n,\epsilon}} \|\overline{\Sigma}_w - I\| \leq 1 + \min_{w \in \Delta_{n,\epsilon}} \|\overline{\Sigma}_w\|$$
$$= 1 + \min_{w \in \Delta_{n,\epsilon}} \max_{M \in \mathcal{M}} \langle \sum_{i=1}^{n} w_i (x_i - \mu)(x_i - \mu)^T, M \rangle$$
$$= 1 + \max_{M \in \mathcal{M}} \min_{w \in \Delta_{n,\epsilon}} \langle \sum_{i=1}^{n} w_i (x_i - \mu)(x_i - \mu)^T, M \rangle. \tag{4}$$

This dual reformulation plays a fundamental role in our analysis. Lemma 2.2 below, proved in Appendix D.2, states that, with high probability, all the terms in the dual reformulation are bounded.

**Lemma 2.2.** *Let $x_1, \ldots, x_n$ be $n$ i.i.d. points from a distribution in $\mathbb{R}^d$ with mean $\mu$ and covariance $\Sigma \preceq I$. Let $Q = \Theta(1/\sqrt{\epsilon} + (1/\epsilon)\sqrt{\mathrm{tr}(\Sigma)/n})$. For $M \in \mathcal{M}$, let $S_M = \{i \in [n] : (x_i - \mu)^T M (x_i - \mu) \leq Q^2\}$. Let $\mathcal{E}$ be the event $\mathcal{E} = \{\forall M \in \mathcal{M}, |S_M| \geq (1 - \epsilon)n\}$. There exists a constant $c > 0$ such that the event $\mathcal{E}$ happens with probability at least $1 - \exp(-c\epsilon n)$.*

Lemma 2.2 draws on the results by Lugosi and Mendelson [LM19b, Proposition 1] and Depersin and Lecue [DL19, Proposition 1]. The proof is given in Appendix D. Importantly, given $n = \Omega(\mathrm{tr}(\Sigma)/\epsilon)$ samples, the threshold $Q$ is $O(1/\sqrt{\epsilon})$. Approximating the empirical process in Eq. (4) with a truncated process allows us to use the powerful inequality for concentration of bounded empirical processes due to Talagrand [Tal96]. Formally, we show the following lemma:

**Lemma 2.3.** *Let $x_1, \ldots, x_n$ be $n$ i.i.d. points from a distribution in $\mathbb{R}^d$ with mean $\mu$ and covariance $\Sigma \preceq I$. Further assume that for each $i$, $\|x_i - \mu\| = O(\sqrt{\mathrm{tr}(\Sigma)/\epsilon})$. There exists $c, c' > 0$ such that for $\epsilon \in (0, c')$, with probability $1 - 2\exp(-cn\epsilon)$, we have that $\min_{w \in \Delta_{n,\epsilon}} \|\overline{\Sigma}_w - I\| \leq \delta^2/\epsilon$, where $\delta = O(\sqrt{(\mathrm{tr}(\Sigma)\log \mathrm{r}(\Sigma))/n} + \sqrt{\epsilon})$.*

*Proof.* Throughout the proof, assume that the event $\mathcal{E}$ from Lemma 2.2 holds. Without loss of generality, also assume that $\mu = 0$. Let $f : \mathbb{R}_+ \to \mathbb{R}_+$ be the following function:

$$f(x) := \begin{cases} x, & \text{if } x \leq Q^2 \\ Q^2, & \text{otherwise.} \end{cases} \tag{5}$$

It follows directly that $f$ is 1-Lipschitz and $0 \leq f(x) \leq x$. Using minimax duality,

$$\min_{w \in \Delta_{n,\epsilon}} \|\overline{\Sigma}_w - I\| \leq 1 + \max_{M \in \mathcal{M}} \min_{w \in \Delta_{n,\epsilon}} \sum w_i x_i^T M x_i \leq 1 + \max_{M \in \mathcal{M}} \sum_{i=1}^{n} f(x_i^T M x_i)/((1 - \epsilon)n),$$

where the second inequality uses that on event $\mathcal{E}$, for every $M \in \mathcal{M}$, the set $S_M = \{[i] \in n : x_i^T M x_i \leq Q^2\}$ has cardinality larger than $(1 - \epsilon)n$, and thus, the uniform distribution on the set $S_M$ belongs to $\Delta_{n,\epsilon}$. Define the following empirical processes $R$ and $R'$:

$$R = \sup_{M \in \mathcal{M}} \sum_{i=1}^{n} f(x_i^T M x_i), \qquad R' = \sup_{M \in \mathcal{M}} \sum_{i=1}^{n} f(x_i^T M x_i) - \mathbb{E}f(x_i^T M x_i).$$

As $0 \leq f(x) \leq x$, we have that $0 \leq \mathbb{E}f(x_i^T M x) \leq \mathbb{E}x_i^T M x \leq 1$, which gives that $|R - R'| \leq n$. Overall, we obtain the following bound:

$$\min_{w \in \Delta_{n,\epsilon}} \|\overline{\Sigma}_w - I\| \leq 1 + R/((1 - \epsilon)n) \leq 1 + 2(R' + n\epsilon)/n \leq (2R')/n + 3.$$

Note that $3 \leq \delta^2/\epsilon$ when $\delta \geq \sqrt{3\epsilon}$. We now apply Talagrand's concentration inequality on $R'$, as each term is bounded by $Q^2$ (see Appendix E.3 for details). Applying the inequality, we get $R'/n = O(\delta^2/\epsilon)$ with probability $1 - \exp(-cn\epsilon)$. By taking a union bound, we get that both $R'/n = O(\delta^2/\epsilon)$ and $\mathcal{E}$ hold with high probability. $\square$

## 2.2 Controlling the Mean

Suppose $u^* \in \Delta_{n,\epsilon}$ achieves the minimum in Lemma 2.3, i.e., $\|\overline{\Sigma}_{u^*} - I\| \leq \delta^2/\epsilon$. It is not necessary that $\|\mu_{u^*} - \mu\| \leq \delta$. Recall that our aim is to find a $w \in \Delta_{n,\epsilon}$ that satisfies the conditions: (i) $\|\mu_w - \mu\| \leq \delta$, and (ii) $\|\overline{\Sigma}_w - I\| \leq \delta^2/\epsilon$. Given $u^*$, we will remove additional $O(\epsilon)$-fraction of probability mass from $u^*$ to obtain a $w \in \Delta_n$ such that $\|\mu_w - \mu\| \leq \delta$. For $u \in \Delta_n$, consider the following set of distributions:

$$\Delta_{n,\epsilon,u} = \left\{ w : \sum_{i=1}^{n} w_i = 1, w_i \leq u_i/(1-\epsilon) \right\}.$$

For any $w \in \Delta_{n,\epsilon,u^*}$, we directly obtain that $\Sigma_w \preceq \Sigma_{u^*}/(1-\epsilon)$. Our main result in this subsection is that, with high probability, there exists a $w^* \in \Delta_{n,4\epsilon,u^*}$ such that $\|\mu_{w^*} - \mu\| \leq \delta$. We first prove an intermediate result, Claim 2.4 below, that uses the truncation (Lemma 2.2) and simplifies the constraint $\Delta_{n,4\epsilon,u^*}$. Let $g : \mathbb{R} \to \mathbb{R}$ be the following thresholding function:

$$g(x) = \begin{cases} x, & \text{if } x \in [-Q,Q], \\ Q, & \text{if } x > Q, \\ -Q, & \text{if } x < -Q. \end{cases} \qquad (6)$$

**Claim 2.4.** *Let $u \in \Delta_{n,\epsilon}$ for $\epsilon < 1/8$. On the event $\mathcal{E}$ (defined in Lemma 2.2), for every $v \in \mathcal{S}^{d-1}$, we have that $\min_{w \in \Delta_{n,4\epsilon,u}} \left| \sum_{i=1}^{n} w_i v^T (x_i - \mu) \right| \leq 5\epsilon Q + 2|\sum_{i \in [n]} g(v^T(x_i - \mu))|/n$.*

Please see Appendix D.3 for the proof. Using Claim 2.4, we prove the following:

**Lemma 2.5.** *Let $x_1, \ldots, x_n$ be $n$ i.i.d. points from a distribution in $\mathbb{R}^d$ with mean $\mu$ and covariance $\Sigma \preceq I$. Let $0 < \epsilon < 1/2$ and $u \in \Delta_{n,\epsilon}$. Then, for a constant $c > 0$, the following holds with probability $1 - \exp(-cn\epsilon)$: $\min_{w \in \Delta_{n,4\epsilon,u}} \|\mu_w - \mu\| \leq \delta$, where $\delta = O\left(\sqrt{\epsilon} + \sqrt{\text{tr}(\Sigma)/n}\right)$.*

At a high-level, the proof of Lemma 2.5 proceeds as follows: We use duality and the variational characterization of the $\ell_2$ norm to reduce our problem to an empirical process over projections. We then use Claim 2.4 to simplify the domain constraint $\Delta_{n,4\epsilon,u^*}$ and obtain a bounded empirical process, with an overhead of $O(\epsilon Q) = O(\delta)$.

*Proof.* (Proof of Lemma 2.5) Let $\Delta$ be the set $\Delta_{n,4\epsilon,u}$ and assume that $\mu = 0$ without loss of generality. On the event $\mathcal{E}$ (defined in Lemma 2.2), using minimax duality and Claim 2.4, we get

$$\min_{w \in \Delta} \max_{v \in \mathcal{S}^{d-1}} \sum_{i=1}^{n} w_i x_i^T v = \max_{v \in \mathcal{S}^{d-1}} \min_{w \in \Delta} \sum_{i=1}^{n} w_i x_i^T v \leq 5\epsilon Q + \max_{v \in \mathcal{S}^{d-1}} | \sum_{i \in [n]} 2g(v^T x_i)/n |. \quad (7)$$

We define the following empirical processes:

$$N = \sup_{v \in \mathcal{S}^{d-1}} \sum_{i=1}^{n} g(v^T x_i), \qquad N' = \sup_{v \in \mathcal{S}^{d-1}} \sum_{i=1}^{n} g(v^T x_i) - \mathbb{E}[g(v^T x_i)].$$

As $g(\cdot)$ is an odd function and $\mathcal{S}^{d-1}$ is an even set, we get that both $N$ and $N'$ are non-negative. For any $v \in \mathcal{S}^{d-1}$, note that $v^T x$ has variance at most 1 and $\mathbb{P}(|v^T x| \geq Q) = O(\epsilon)$. We can thus bound $\mathbb{E}g(v^T x)$ as $O(\sqrt{\epsilon}) = O(\epsilon Q)$ (see Proposition C.3). This gives us that $|N - N'| = O(n\epsilon Q)$. Using the variational form of the $\ell_2$ norm with Eq. (7) leads to the following inequality in terms of $N'$:

$$\min_{w \in \Delta} \|\mu_w\| = \max_{v \in \mathcal{S}^{d-1}} \min_{w \in \Delta} \sum_{i=1}^{n} w_i x_i^T v \leq 5\epsilon Q + 2N/n) = O(\epsilon Q) + (2N')/n .$$

Note that the term $\epsilon Q$ is small as $\epsilon Q = O(\delta)$. As $N'$ is a bounded empirical process, with the bound $Q$, we can apply Talagrand's concentration inequality. We defer the details to Appendix E, showing that $N'/n = O(\sqrt{\text{tr}\,\Sigma/n} + \sqrt{\epsilon}) = O(\delta)$. Taking a union bound over concentration of $N'$ and the event $\mathcal{E}$, we get that the desired result holds with high probability. $\qquad \square$

**Proof Sketch of Theorem 1.4** By Lemma 2.3, we get that there exists a $u^* \in \Delta_{n,\epsilon}$ such that $\|\overline{\Sigma}_{u^*} - I\| \leq \delta^2/\epsilon$. Applying Lemma 2.5 with this $u^*$, we get that there exists a $w^* \in \Delta_{n,4\epsilon,u^*}$ such that $\|\mu_{u^*} - \mu\| \leq \delta$. $v^T \overline{\Sigma}_{w^*} v \leq (1/(1-4\epsilon))v^T \overline{\Sigma}_{u^*} v = O(\delta^2/\epsilon)$, for small enough $\epsilon$. To obtain a discrete set, we show that rounding $w^*$ to a discrete set only leads to slightly worse constants. We refer the reader to Appendix E.5 for the formal proof of the statement.

## 3 Robust Mean Estimation Under Bounded Central Moments

We sketch the arguments for the proof of Theorem 1.8, which obtains a tighter dependence on $\epsilon$. In the bounded covariance setting, we considered $\delta$ such that $\delta = \Omega(\sqrt{\epsilon})$. As such, we only needed an upper bound on second moment matrix, $\overline{\Sigma}_{S'}$, for a set $S' \subseteq S$ (For $\delta \geq \sqrt{\epsilon}$, the lower bound in the second condition of stability is trivial). For $\delta = o(\sqrt{\epsilon})$, we need a *sharp* lower bound on the minimum eigenvalue of $\overline{\Sigma}_{S_1}$ for *all large subsets* $S_1$ of a set $S'$. Such a result is not possible in general, unless we impose both: (i) identity covariance and (ii) tighter control on tails of $X$. For identity covariance, and bounded central moments, we prove the following:

**Lemma 3.1.** *(informal) Consider the setting of Theorem 1.8. Then, with probability $1 - \tau$, we have that* $\min_{S':|S'|\geq(1-\epsilon)n} \min_{v \in \mathcal{S}^{d-1}} v^T \overline{\Sigma}_{S'} v \geq 1 - \delta^2/\epsilon$*, where $\delta$ is defined in Theorem 1.8.*

See Appendix F for a precise statement. Lemma 3.1 is similar in spirit to [KM15, Theorem 1.3], that only controls the minimum eigenvalue of the set $S$. Given Lemma 3.1, our proof technique follows the same strategy as before: (i) minimax duality, (ii) truncation, and (iii) concentration of bounded empirical processes. We also need a tighter bound on truncation threshold, than the statement of Lemma 2.2, which has a better dependence on $\epsilon$.

**Lemma 3.2.** *Consider the setting in Theorem 1.8. Let $Q_k = \Theta(\sigma_k \epsilon^{-1/k} + (1/\epsilon)\sqrt{d/n})$. There exists a $c > 0$ such that with probability at least $1 - \exp(-c\epsilon n)$, $\sup_{M \in \mathcal{M}} \left| \{ i : x_i^T M x_i \geq Q_k^2 \} \right| \leq \epsilon n$.*

We refer the reader to Appendix F for the detailed proof.

## 4 Conclusions and Open Problems

In this paper, we showed that a standard stability condition from the recent high-dimensional robust statistics literature suffices to obtain near-subgaussian rates for robust mean estimation in the strong contamination model. With a simple pre-processing (bucketing), this leads to efficient outlier-robust estimators with subgaussian rates under only a bounded covariance assumption. An interesting technical question is whether the extra $\log d$ factor in Theorem 1.4 is actually needed. (Our results imply that it is not needed when $\epsilon = \Omega(1)$.) If not, this would imply that stability-based algorithms achieve subgaussian rates without the pre-processing.

## Acknowledgments and Disclosure of Funding

ID was supported by NSF Award CCF-1652862 (CAREER) and a Sloan Research Fellowship. DK was supported by NSF Award CCF-1553288 (CAREER) and a Sloan Research Fellowship. AP was supported by the UW-Madison Institute for Foundations of Data Science (IFDS), NSF grant CCF-1740707.

## Broader Impact

Our work fits within the area of algorithmic high-dimensional robust statistics and aims to advance the algorithmic foundations of outlier-robust learning for heavy-tailed data. An important motivation for this line of work is to design provable defenses of machine learning systems against data poisoning with optimal accuracy-confidence tradeoffs.

As the primary focus of our work is theoretical, we do not expect our results to have immediate societal impact. Nonetheless, we believe that our probabilistic analysis provides useful insights that could be leveraged in practically relevant robust estimators.

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
