[Supplementary Material]

# Supplementary Material

**Additional Notation** For a set $E$, we use $\mathbb{I}(x \in E)$ to denote the indicator function for event $E$. For simplicity, we use $\mathbb{I}(x \geq t)$ to denote the indicator function for the event $E = \{x : x \geq t\}$. For a random variable $Z$, we use $\mathbb{V}(Z)$ to denote its variance. We use $d_{\mathrm{TV}}(p, q)$ to denote the total variation distance between distributions $p$ and $q$. For a vector $w \in \mathbb{R}^n$, we use $\|w\|_1$ to denotes its $\ell_1$ norm, i.e., $\|w\|_1 = \sum_{i=1}^{n} |w_i|$.

## A  Robust Mean Estimation and Stability

### A.1  Robust Mean Estimation from Subset Stability

The theorem statement in [DK19, Theorem 2.7] requires that the input multiset $S$ is stable. We note that the arguments straightforwardly go through when $S$ contains a large stable subset $S' \subseteq S$ (see, e.g., [DKK$^+$16, DKK$^+$17, DHL19]).

For concreteness, we describe a simple pre-processing of the data, that ensures that the data follows the definition as is: simply throw away points so that the cardinality of the corrupted set matches the cardinality of the stable subset.

**Proposition A.1.** *Let $S$ be a set such that $\exists S' \subseteq S$ such that $|S'| \geq (1-\epsilon)|S|$ and $S'$ is $(C\epsilon, \delta)$ for some $C > 0$. Let $T$ be an $\epsilon$-corrupted version of $S$. Let $T'$ be the multiset obtained by removing $\epsilon n$ points of $T$. Let $\epsilon' = \frac{2\epsilon}{1-\epsilon}$. Then $T'$ is an $\epsilon'$-corrupted version of a $((C-1)\epsilon'/2, \delta)$ stable set.*

*Proof.* Let $T$ be an $\epsilon$-corrupted version of $S$. That is, $T = S \cup A \setminus R$. We now remove $\epsilon n$ points arbitrarily from $T$ to obtain the multiset $T'$ of cardinality $(1-\epsilon)n$.

Let $S_2$ be any subset of $S'$ such that $|S_2| = |T_1| = (1-\epsilon)n$. Therefore, $T'$ is at most $(2\epsilon)/(1-\epsilon)$-corrupted version of $S_2$. As $S'$ is $(C\epsilon, \delta)$ stable and $S_2$ is a large subset of $S'$, Claim A.2 states that $S_2$ is $(\epsilon_2, \delta)$ stable where $\epsilon_2 \geq 1 - (1 - C\epsilon)/(1-\epsilon) = (C-1)\epsilon'/2$. $\qquad\square$

**Claim A.2.** *If a set $S$ is $(\epsilon, \delta)$ stable, then its subset $S'$ of cardinality $m > (1-\epsilon)n$ is $(1-(1-\epsilon)\frac{n}{m}, \delta)$ stable.*

*Proof.* To show that $S'$ is $(\epsilon', \delta)$ stable, it suffices to ensure that $\epsilon' \leq \epsilon$ and $(1-\epsilon')|S'| \geq (1-\epsilon)|S|$. Therefore, we require that

$$(1-\epsilon')m \geq (1-\epsilon)n \implies \epsilon' \leq 1 - \frac{(1-\epsilon)n}{m}.$$

The upper bound is always less than $\epsilon$ for $m \leq n$. $\qquad\square$

### A.2  Adapting to Unknown Upper Bound on Covariance

As stated, the stability-based algorithms in [DKK$^+$17, DK19] assume that the inliers are drawn from a distribution with unknown bounded covariance $\Sigma \preceq \sigma^2 I$, where the parameter $\sigma > 0$ is known. Here we note that essentially the same algorithms work even if the parameter $\sigma > 0$ is unknown. For this, we establish the following simple modification of standard results, see, e.g., [DK19].

**Theorem A.3.** *Let $T \subset \mathbb{R}^d$ be an $\epsilon$-corrupted version of a set $S$, where $S$ is $(C\epsilon, \delta)$-stable with respect to $\mu_S$ and $\sigma^2$, where $C > 0$ is a sufficiently large constant. There exists a polynomial time algorithm that given $T$ and $\epsilon$ (but not $\sigma$ or $\delta$) returns a vector $\widehat{\mu}$ so that $\|\mu_S - \widehat{\mu}\| = O(\sigma\delta)$.*

*Proof.* The algorithm is very similar to the algorithm from [DK19] except for the stopping condition. We define a weight function $w : T \to \mathbb{R}_{\geq 0}$ initialized so that $w(x) = 1/|T|$ for all $x \in T$. We iteratively do the following:

- Compute $\mu(w) = \frac{1}{\|w\|_1} \sum_{x \in T} w(x)x$.

- Compute $\Sigma(w) = \frac{1}{\|w\|_1} \sum_{x \in T} w(x)(x - \mu(w))(x - \mu(w))^T$.

- Compute an approximate largest eigenvector $v$ of $\Sigma(w)$.

- Define $g(x)$ for $x \in T$ as $g(x) = |v \cdot (x - \mu(w))|^2$.

- Find the largest $t$ so that $\sum_{x \in T: g(x) \geq t} w(x) \geq \epsilon$.

- Define $f(x) = \begin{cases} g(x) & \text{if } g(x) \geq t \\ 0 & \text{otherwise} \end{cases}$.

- Let $m$ be the largest value of $f(x)$ for any $x \in T$ with $w(x) \neq 0$.

- Set $w(x)$ to $w(x)(1 - f(x)/m)$ for all $x \in T$.

We then repeat this loop unless $\|w\|_1 < 1 - 2\epsilon$, in which case we return $\mu(w)$.

Note that if $S$ is $(\epsilon, \delta)$-stable with respct to $\mu_S$ and $\sigma^2$, then $S/\sigma$ is $(\epsilon, \delta)$ with respect to $\mu_S/\sigma$ and $1$. We note that if $\sigma$ was known, the weighted universal filter algorithm of [DK19] could be applied to $T/\sigma$ in order to learn $\mu_S/\sigma$ to error $O(\delta)$. Multiplying the result by $\sigma$ would yield an approximation to $\mu_S$ with error $O(\sigma\delta)$. We note that this algorithm is equivalent to the one provided above, except that we would stop the loop as soon as $\Sigma(w) \leq \sigma(1 + O(\delta^2/\epsilon))$ rather than waiting until $\|w\|_1 \leq 1 - 2\epsilon$.

However, we note that by the analysis in [DK19] of this algorithm, that at each iteration until it stops, $\sum_{x \in S} w(x)$ decreases by less than $\sum_{x \in T \setminus S} w(x)$ does. Since the latter cannot decrease by more than $\epsilon$, this means that the algorithm of [DK19] would stop before ours does. Our algorithm then continues to remove an additional $O(\epsilon)$ mass from the weight function $w$ (but only this much since $f$ has support on points of mass only a bit more than $\epsilon$). It is easy to see that these extra removals do not increase $\Sigma(w)$ by more than a factor of $1 + O(\epsilon)$. This means that when our algorithm terminates $\Sigma(w)/\sigma \leq I + O(\delta^2/\epsilon)$. Thus, by the weighted version of Lemma 2.4 of [DK19], we have that

$$\|\mu_S - \mu(w)\| = \sigma\|\mu_S/\sigma - \mu(w)/\sigma\| \leq \sigma O(\delta + \sqrt{\epsilon(\delta^2/\epsilon)}) = O(\sigma\delta) .$$

This completes the proof. $\qquad\square$

## B  Robust Mean Estimation using Median-of-Means Principle

In this section, we again consider distributions with finite covariance matrix $\Sigma$. We now turn our attention to the proof of Theorem 1.7 that removes the additional logarithmic factor $\sqrt{\log(r(\Sigma))}$. In Section B.1, we show a result stating that pre-processing on i.i.d. points yields a set that contains a large stable subset (after rescaling). Then, in Section B.2, we use a coupling argument to show a similar result in the strong contamination model.

We recall the median of means principle. Let $k \in [n]$.

1. First randomly bucket the data into $k$ disjoint buckets of equal size (if $k$ does not divide $n$, remove some samples) and compute their empirical means $z_1, \ldots, z_k$.

2. Output (appropriately defined) multivariate median of $z_1, \ldots, z_k$.

### B.1  Stability of Uncorrupted Data

We first recall the result (with different constants) from Depersin and Lecué [DL19] in a slightly different notation.

**Theorem B.1.** *[DL19, Proposition 1] Let $z_1, \ldots, z_k$ be $k$ points in $\mathbb{R}^d$ obtained by the median-of-means preprocessing on $n$ i.i.d. data $x_1, \ldots, x_n$ from a distribution with mean $\mu$ and covariance $\Sigma$. Let $\mathcal{M}$ be the set of PSD matrices with trace at most $1$. Then, there exists a constant $c > 0$, such that with probability at least $1 - \exp(-ck)$, we have that for all $M \in \mathcal{M}$, $\left|\{i \in [k] : (z_i - \mu)^T M(z_i - \mu) > (k\|\Sigma\|/n)\delta^2\}\right| \leq \frac{k}{100}$, where $\delta = O(\sqrt{r(\Sigma)/k} + 1)$.*

We now state our main result in this section, proved using minimax duality, that Theorem B.1 implies stability. We first consider the case of i.i.d. data points, as it conveys the underlying idea clearly.

**Theorem B.2.** *Let $x_1, \ldots, x_n$ be $n$ i.i.d. random variables from a distribution with mean $\mu$ and covariance $\Sigma \preceq I$. For $k \in [n]$, let $z_1, \ldots, z_k$ be the variables obtained by median-of-means*

*preprocessing. Then, with probability $1 - \exp(-ck)$, where $c$ is a positive universal constant, there exists a set $S_1 \subseteq [k]$ and $|S_1| \geq 0.95k$ such that $S_1$ is $(0.1, \delta)$-stable with respect to $\mu$ and $k\|\Sigma\|/n$, where $\delta = O(\sqrt{r(\Sigma)/n} + 1)$.*

*Proof.* For brevity, let $\sigma = \sqrt{k\|\Sigma\|/n}$. Suppose that the conclusion in Theorem B.1 holds with $\delta = O(\sqrt{r(\Sigma)/k} + 1)$ such that $\delta \geq 1$, i.e., for every $M \in \mathcal{M}$, for at least $0.99k$ points $(z_i - \mu)^T M(z_i - \mu) \leq \sigma^2 \delta^2$. Using minimax duality, we get that

$$\min_{w \in \Delta_{k,0.01}} \left\| \sum_{i=1}^{k} w_i(z_i - \mu)(z_i - \mu)^T \right\| = \min_{w \in \Delta_{k,0.01}} \max_{M \in \mathcal{M}} \langle M, \sum_{i=1}^{k} w_i(z_i - \mu)(z_i - \mu)^T \rangle$$

$$= \max_{M \in \mathcal{M}} \min_{w \in \Delta_{k,0.01}} \langle M, \sum_{i=1}^{k} w_i(z_i - \mu)(z_i - \mu)^T \rangle$$

$$\leq \sigma^2 \delta^2,$$

where the last step uses the conclusion of Theorem B.1. As $\delta^2 \geq 1$, we also get that $\| \sum_{i=1}^{k} w_i^*(z_i - \mu)(z_i - \mu)^T - \sigma^2 I \| \leq \sigma^2 \delta^2$. Let $w^*$ be the distribution that achieves the minimum in the above statement. We can also bound the first moment of $w^*$ using the bound on the second moment of $w^*$ as follows:

$$\sum_{i=1}^{k} w_i^* v^T(z_i - \mu) \leq \sqrt{\sum_{i=1}^{k} w_i^*(v^T(z_i - \mu))^2} \leq \sqrt{\| \sum_i w_i^*(z_i - \mu)(z_i - \mu)^T \|} \leq \sqrt{\sigma^2 \delta^2} = \sigma \delta.$$

Given this $w^* \in \Delta_{k,0.01}$, we will now obtain a subset of $\{z_1, \ldots, z_k\}$ that satisfies the stability condition. In particular, Lemma E.2 shows that we can deterministically round $w^*$ such that there exists a large stable subset of $\{z_1, \ldots, z_k\}$ which is $(0.1, \delta)$ stable with respect to $\mu$ and $\sigma^2$. □

## B.2  Stability Under Strong Contamination Model

We now prove Theorem 1.7, i.e., stability of a subset after corruption, using Theorem B.2. The following result shares the same principle as [DHL19, Lemma B.1]: we add a coupling argument because the pre-processing step (random bucketing) introduces an additional source of randomness.

**Theorem B.3.** *(Formal statement of Theorem 1.7) Let $T$ be an $\epsilon$-corrupted version of the set $S$, where $S$ is a set of $n$ i.i.d. points from a distribution $P$ with mean $\mu$ and covariance $\Sigma$. Set $\epsilon' = O(\epsilon + \log(1/\tau)/n)$ and set $k = \lfloor \epsilon'n \rfloor$. Let $T_k$ be the set of $k$ points obtained by median-of-means preprocessing on the set $T$. Then, with probability $1 - \tau$, $T_k$ is $0.01$-corruption of a set $S_k$ such that there exists a $S_k' \subseteq S_k$, $|S_k'| \geq 0.95k$ and $S_k'$ is $(0.1, \delta)$ stable with respect to $\mu$ and $k\|\Sigma\|/n$, where $\delta = O(\sqrt{r(\Sigma)/n} + 1)$.*

*Proof.* For simplicity, assume $k$ divides $n$ and let $m = n/k$.

Let $S = \{x_1, \ldots, x_n\}$ be the multiset of $n$ i.i.d. points in $\mathbb{R}^d$ from $P$. We can write $T$ as $T = \{x_1', \ldots, x_n'\}$ such that $|\{i : x_i' \neq x_i\}| \leq \epsilon n$.

As the algorithm only gets a multiset, we first order them arbitrarily. Let $r_1', \ldots, r_n'$ be any arbitrary labelling of points and let $\sigma_1(\cdot)$ be the permutation such that $r_i' = x_{\sigma_1(i)}'$. We now split the points randomly into buckets by randomly shuffling them. Let $\sigma(\cdot)$ be a uniformly random permutation of $[n]$ independent of $T$ (and $S$). Define $w_i' = r_{\sigma(i)}' = x_{\sigma_1(\sigma(i))}'$. For $i \in [k]$, define the bucket $B_i'$ to be the multiset $B_i' := \{w_{(i-1)m+1}', \ldots, w_{im}'\}$. For $i \in [k]$, define $z_i'$ to be the mean of the set $B_i'$, i.e., $z_i = \mu_{B_i'}$. That is, the input to the stable algorithm would be the multiset $T_k$, where $T_k = \{z_1', \ldots, z_k'\}$.

We now couple the corrupted points with the original points. For $\sigma$ and $\sigma_1$, define their composition $\sigma'$ as $\sigma'(i) := \sigma_1(\sigma(i))$. Define $r_i := x_{\sigma_1(i)}$ and $w_i := r_{\sigma(i)} = x_{\sigma'(i)}$. Importantly, Proposition B.4 below states that $w_i$'s are i.i.d. from $P$. The analogous bucket for uncorrupted samples is $B_i := \{w_{(i-1)m+1}, \ldots, w_{im}\}$. For $i \in [k]$, define $z_i := \mu_{B_i}$ and define $S_k$ to be $\{z_1, \ldots, z_k\}$. Therefore,

$z_1, \ldots, z_k$ are obtained from the median-of-means processing of i.i.d. data $w_1, \ldots, w_n$, and thus Theorem B.2 holds[1]. That is, there exists $S'_k \subseteq S_k$ that satisfies the desired properties.

It remains to show that $T_k$ is a corruption of $S_k$. It is easy to see that $|T_k \cap S_k| \geq k - \epsilon n \geq 0.99k$, by choosing $\epsilon'$ large enough. That is, for any $\sigma_1$ and $\sigma$, $T_k$ is at most $(0.01)$-contamination of the set $S_k$. $\qquad\square$

**Proposition B.4.** *Let $x_1, \ldots, x_n$ be $n$ i.i.d. points from a distribution $P$ and $\sigma_1(\cdot)$ be a permutation, potentially depending on $x_1, \ldots, x_n$. Let $\sigma(\cdot)$ be a random permutation independent of $x_1, \ldots, x_n$ and $\sigma_1(\cdot)$. Define the composition permutation be $\sigma'(i) := \sigma_1(\sigma(i))$. Then $x_{\sigma'(1)}, \ldots, x_{\sigma'(n)}$ are also i.i.d. from the distribution $P$.*

*Proof.* First observe that $\sigma'(\cdot)$ is a uniform random permutation independent of $x_1, \ldots, x_n$. The result follows from the following fact:

**Fact B.5.** *Let $x_1, \ldots, x_n$ be $n$ i.i.d. points from a distribution $P$. Let $\sigma(\cdot)$ be a random permutation independent of $x_1, \ldots, x_n$, then $x_{\sigma(1)}, \ldots, x_{\sigma(n)}$ are also i.i.d. from the distribution $P$.*

$\qquad\square$

# C  Tools from Concentration and Truncation

**Organization.** In Section C.1, we state the concentration results that we will use repeatedly in the following sections. Section C.2 contains some well-known results regarding the properties of the truncated distribution.

## C.1  Concentration Results

We first state Talagrand's concentration inequality for bounded empirical processes.

**Theorem C.1.** *([BLM13, Theorem 12.5] ) Let $Y_1, \ldots, Y_n$ be independent identically distributed random vectors. Assume that $\mathbb{E}Y_{i,s} = 0$, and that $Y_{i,s} \leq L$ for all $s \in \mathcal{T}$. Define*

$$Z = \sup_{s \in \mathcal{T}} \sum_{i=1}^n Y_{i,s}, \qquad \sigma^2 = \sup_{s \in \mathcal{T}} \sum_{i=1}^n \mathbb{E}Y_{i,s}^2.$$

*Then, with probability at least $1 - \exp(-t)$, we have that*

$$Z = O(\mathbb{E}Z + \sigma\sqrt{t} + Lt). \tag{8}$$

*See [BLM13, Exercise 12.15] for explicit constants.*

We will also repeatedly use the following version of Matrix Bernstein inequality [Tro15, Min17].

**Theorem C.2.** *([Tro15, Corollary 7.3.2]) Let $S_1, \ldots, S_n$ be $n$ independent symmetric matrices such that $\mathbb{E}S_i = 0$ and $\|S_i\| \leq L$ a.s. for each index $i$. Let $Z = \sum_{i=1}^n S_i$ and let $V$ be any PSD matrix such that $\sum_{i=1}^n \mathbb{E}S_k S_k^T \preceq V$. Let $\nu = \|V\|$ and $r = \mathrm{r}(V)$. Then, we have that*

$$\mathbb{E}\|Z\| = O(\sqrt{\nu \log r} + L \log r). \tag{9}$$

*In particular, if $S_i = \xi_i x_i x_i^T$, where $\xi_i$ is a Rademacher random variable, and $x_i$ is sampled independently from a distribution with zero mean, covariance $\Sigma$, and bounded support $L$, i.e. $\|x_i\| \leq L$ almost surely. Then $\mathbb{E}\|Z\| = O(\sqrt{nL\|\Sigma\| \log \mathrm{r}(\Sigma)} + L \log \mathrm{r}(\Sigma))$.*

## C.2 Properties under Truncation

We state some basic results regarding truncation of a distribution in this subsection. These results are well-known in literature and are included here for completeness (see, e.g., [DKK+17, LRV16]).

**Proposition C.3.** *(Shift in mean by truncation) Let $X$ be sampled from a distribution with mean $0$ and covariance $\Sigma \preceq I$. For a $t \geq 0$, let $g(\cdot)$ be defined as*

$$g(x) = \begin{cases} x, & \text{if } x \in [-t, t], \\ t, & \text{if } x > t, \\ -t, & \text{if } x < -t. \end{cases}$$

*If $t \geq C\epsilon^{-\frac{1}{2}}$, then for all $v \in \mathcal{S}^{d-1}$, $|\mathbb{E}g(x^T v)| \leq C^{-1}\sqrt{\epsilon}$.*

*Proof.* Let $Z = x^T v$. By Markov's inequality,

$$\mathbb{P}(Z \geq t) \leq \mathbb{P}(Z^2 \geq C^2 \epsilon^{-1}) \leq \frac{1}{C^2 \epsilon^{-1}} = C^{-2}\epsilon.$$

We get that

$$|\mathbb{E}g(Z)| = |\mathbb{E}Z - g(Z)| \leq \mathbb{E}|Z - g(Z)| \leq \mathbb{E}|Z|\mathbb{I}_{|Z|\geq t} \leq \sqrt{\epsilon}C^{-1}. \tag{10}$$

$\square$

**Proposition C.4.** *(Shift in mean by truncation under higher moments) Let $X$ be sampled from a distribution with mean $0$ and covariance $(1 - \sigma_k^2 \epsilon^{1-\frac{2}{k}})I \preceq \Sigma \preceq I$. Moreover, assume that the distribution has bounded moments, i.e., for a $k \geq 4$:*

$$\forall v \in \mathcal{S}^{d-1}, \quad (\mathbb{E}(v^T X)^k)^{\frac{1}{k}} \leq \sigma_k. \tag{11}$$

*Note that $\sigma_2 \leq 1$. Let $T_k = \sigma_k \epsilon^{-\frac{1}{k}}$. Then*

1. *For all $M \in \mathcal{M}$, $\mathbb{E}(x^T M x)^{\frac{k}{2}} \leq \sigma_k^k$.*

2. *For all $M \in \mathcal{M}$ and $t \geq CT_k^2$, $\mathbb{E}x^T M x \mathbb{I}_{x^T M x \geq t} \leq \sigma_k^2 C^{\frac{2}{k}-1} \epsilon^{1-\frac{2}{k}}$.*

3. *Let $f(\cdot)$ be defined as $f(x) = \min(x, t)$. For a $t \geq CT_k^2$, $|\mathbb{E}f(x^T M x) - 1| \leq \sigma_k^2 \epsilon^{1-\frac{2}{k}}(1 + C^{1-\frac{k}{2}})$.*

4. *Let $t \geq CT_k$. For all $v \in \mathcal{S}^{d-1}$, $|\mathbb{E}x^T v \mathbb{I}_{|x^T v| \leq t}| \leq \sigma_k \epsilon^{1-\frac{1}{k}} C^{1-k}$.*

5. *Let $g(\cdot)$ be defined as $g(x) = sign(x)\min(|x|, t)$. For $t \geq CT_k$ and all $v \in \mathcal{S}^{d-1}$, $|\mathbb{E}g(x^T v)| \leq \sigma_k C^{1-k} \epsilon^{1-\frac{1}{k}}$.*

6. *$\mathbb{E}\|X\|^k \leq d^{\frac{k}{2}} \sigma_k^k$.*

7. *$\mathbb{P}(\|X\| \geq \sigma_k \sqrt{d} \epsilon^{-1/k}) \leq \epsilon$.*

*Proof.* We prove each statement in turn.

1. We use the spectral decomposition of $M$, to write $M = U^T \Delta U$, where $U$ is a rotation matrix, $\Delta$ is a non-negative diagonal matrix with diagonal entries $\lambda_i$ and trace 1. Observe that if the random variable $X$ satisfies Equation (11), then the random variable $Z := UX$ also satisfies Equation (11).

   We use the aforementioned observation and apply Jensen's inequality to get:

$$\mathbb{E}(x^T M x)^{\frac{k}{2}} = \mathbb{E}(Z^T \Delta Z)^{\frac{k}{2}} = \mathbb{E}(\sum_{i=1}^d \lambda_i z_i^2)^{\frac{k}{2}} \leq \sum_{i=1}^d \lambda_i \mathbb{E}z_i^k \leq \sum_{i=1}^d \lambda_i \sigma_k^k \leq \sigma_k^k.$$

2. Let $Z = x^T M x$. From the first part, we have that $\frac{k}{2}$-th moment of $Z$ is bounded by $\sigma_k^2$. By Markov's inequality, we get that

$$\mathbb{P}\{Z \geq t\} \leq \mathbb{P}\{Z \geq CT_k^2\} \leq \mathbb{P}\left\{Z \geq C\frac{\sigma_k^2}{\epsilon^{\frac{2}{k}}}\right\} \leq \frac{\epsilon}{C^{\frac{k}{2}}\sigma_k^k}\mathbb{E}Z^{\frac{k}{2}} \leq \frac{\epsilon}{C^{\frac{k}{2}}}.$$

We can now apply Hölder's inequality, to get

$$\mathbb{E}Z\mathbb{I}_{Z \geq CT_k^2} \leq \sigma_k^2 C^{\frac{2}{k}-1}\epsilon^{1-\frac{2}{k}}.$$

3. As above, let $Z = x^T M x$. It follows that $f(x) \leq x$. Therefore, we get that

$$\mathbb{E}f(x^T M x) \leq \mathbb{E}x^T M x \leq 1.$$

For the lower bound, we get that

$$\mathbb{E}f(x^T M x) \geq \mathbb{E}x^T M x \mathbb{I}_{x^T M x \leq CT_k^2} = \mathbb{E}x^T M x 1 - \mathbb{E}x^T M x \mathbb{I}_{x^T M x > CT_k^2}$$
$$\geq 1 - \sigma_k^2 \epsilon^{1-\frac{2}{k}} - \sigma_k^2 \epsilon^{1-\frac{2}{k}}C^{1-\frac{k}{2}}.$$

4. Let $Z = x^T v$. We note that

$$\mathbb{P}(Z \geq t) \geq \mathbb{P}(Z \geq CT_k) \leq \mathbb{P}(Z^k \geq C^k T_k^k) \leq \frac{\sigma_k^k}{\sigma_k^k \epsilon^{-1} C^k} \leq C^{-k}\epsilon.$$

We now bound the deviation in mean by truncation:

$$\mathbb{E}Z = \mathbb{E}Z\mathbb{I}_{|Z| \leq t} + \mathbb{E}Z\mathbb{I}_{|Z| > t} = 0$$
$$\implies |\mathbb{E}Z\mathbb{I}_{|Z| \leq t}| = |\mathbb{E}Z\mathbb{I}_{Z > t}|$$
$$\leq (\mathbb{E}Z^k)^{\frac{1}{k}}(\mathbb{P}\{Z > t\})^{1-\frac{1}{k}}$$
$$= \sigma_k C^{1-k}\epsilon^{1-\frac{1}{k}}.$$

5. Let $Z = x^T v$. We get that

$$|\mathbb{E}g(Z)| = |\mathbb{E}Z - g(Z)| \leq \mathbb{E}|Z - g(Z)| \leq \mathbb{E}|Z|\mathbb{I}_{|Z| \geq CT_k} \leq \sigma_k \epsilon^{1-\frac{1}{k}}C^{1-k}.$$

6. It follows by taking $M = \frac{1}{d}I$ in the first part.

7. This follows by Markov's inequality and the previous part.

$\square$

**Lemma C.5.** *Let $P$ be a distribution with mean $\mu$ and covariance $I$. Let $X \sim P$. For $k > 2$, let its $k$-th central moment be bounded as*

$$\text{for all } v \in \mathcal{S}^{d-1}: \quad (\mathbb{E}|v^T X|^k)^{\frac{1}{k}} \leq \sigma_k.$$

*For $\epsilon \leq 0.5$, let $E$ be the event*

$$E = \{\|X - \mu\| \leq T\},$$

*where $T$ is such that $\mathbb{P}(E) \geq 1 - \epsilon$. Let $Z$ be the random variable $X|E$, that is $X$ conditioned on $X \in E$. Then, we have that*

1. $\|\mu - \mathbb{E}Z\| \leq \frac{1}{1-\epsilon}\sigma_k \epsilon^{1-\frac{1}{k}} \leq 2\sigma_k \epsilon^{1-\frac{1}{k}}$.

2. $(1 - 3\sigma_k^2 \epsilon^{1-\frac{2}{k}})I \preceq \mathrm{Cov}(Z) \preceq I$.

*Proof.* We prove each statement in turn.

1. Let $Q$ be the distribution of $Z$. We will assume that $\mathbb{P}(E^c) > 0$, otherwise the results hold trivially. Let $R$ be the distribution of $X$ conditioned on $X \in E^c$ and let $Y \sim R$. Note that $P$ can be written as the convex combination of $Q$ and $R$.

$$P = (\mathbb{P}(E))Q + (1 - \mathbb{P}(E))R. \tag{12}$$

Using this decomposition, we can calculate the shift in mean along any direction $v \in \mathcal{S}^{d-1}$:

$$\mathbb{P}(E)v^T \mathbb{E}Z + (1 - \mathbb{P}E)\mathbb{E}v^T Y = v^T \mathbb{E}X = \mu$$
$$\implies v^T(\mathbb{E}Z - \mu) = \frac{1}{\mathbb{P}(E)}\mathbb{E}(-v^T(X - \mu))\mathbb{I}_{X \notin E}$$
$$\leq \frac{1}{\mathbb{P}(E)}(\mathbb{E}|v^T(X-\mu)|^k)^{\frac{1}{k}}(\mathbb{P}(E^c))^{1-\frac{1}{k}}$$
$$\leq \frac{1}{\mathbb{P}(E)}\sigma_k \epsilon^{1-\frac{1}{k}},$$

   where the first inequality uses Hölder's inequality. Therefore, $\|\mathbb{E}Z - \mu\| \leq \sigma_k \epsilon^{1-1/k}/(1-\epsilon)$.

2. We will follow the notations from the previous part. Note that for all $v \in \mathcal{S}^{d-1}$, the mean minimizes the quadratic loss

$$\mathbb{E}(v^T(Z - \mathbb{E}Z))^2 \leq \mathbb{E}(v^T(Z - \mu))^2.$$

   Note that for any direction $v$, we have that $\mathbb{E}(v^T(Z - \mu))^2 \leq \mathbb{E}(v^T(Y - \mu))^2$. As $\mathbb{E}(v^T(X - \mu))^2$ is the convex combination of $\mathbb{E}(v^T Z)^2$ and $\mathbb{E}(v^T Y)^2$, and thus larger than the minimum of these two, we get

$$\mathbb{E}(v^T(Z - \mu))^2 = \min(\mathbb{E}(v^T(Y - \mu))^2, \mathbb{E}(v^T(Z - \mu))^2)$$
$$\leq \mathbb{P}(E)\mathbb{E}(v^T(Z - \mu))^2 + (1 - \mathbb{P}(E))\mathbb{E}(v^T(Y - \mu))^2 = \mathbb{E}(v^T(X - \mu))^2 = 1.$$

   Therefore, we obtain the following upper bound:

$$\mathbb{E}v^T(Z - \mathbb{E}Z)^2 \leq \mathbb{E}(v^T(Z - \mu))^2 \leq 1.$$

   We now turn our attention to lower bound. We first note that

$$(1 - \mathbb{P}(E))\mathbb{E}(v^T(Y - \mu))^2 = \mathbb{E}(v^T X)^2 \mathbb{I}\{X \in E^c\} \leq (\mathbb{E}(v^T X)^k)^{\frac{2}{k}}(\mathbb{P}(E))^{1-\frac{2}{k}} \leq \sigma_k^2 \epsilon^{1-\frac{2}{k}}.$$

   Using the definition of $P$, $Q$ and $R$, we get

$$\mathbb{E}(v^T(Z - \mu))^2 = \frac{1}{\mathbb{P}(E)}(\mathbb{E}(v^T(X - \mu))^2 - (1 - \mathbb{P}(E))\mathbb{E}(v^T(Y - \mu))^2)$$
$$\geq (1 - (1 - \mathbb{P}(E))\mathbb{E}(v^T(Y - \mu))^2) \geq 1 - \sigma_k^2 \epsilon^{1-\frac{2}{k}}.$$

   We are now ready to bound the lower bound the deviation from mean:

$$\mathbb{E}(v^T(Z - \mathbb{E}Z))^2 = \mathbb{E}(v^T(Z - \mu))^2 - (\mathbb{E}Z - \mu)^2$$
$$\geq 1 - \sigma_k^2 \epsilon^{1-\frac{2}{k}} - (\frac{\sigma_k \epsilon^{1-\frac{1}{k}}}{1 - \epsilon})^2$$
$$\geq 1 - \sigma_k^2 \epsilon^{1-\frac{2}{k}} - \frac{\sigma_k^2 \epsilon^{1-\frac{2}{k}}}{1 - \epsilon} \geq 1 - 3\sigma_k^2 \epsilon^{1-\frac{2}{k}}.$$

$\square$

## D Bounds on the Number of Points with Large Projections

**Organization.** This section contains the proofs of Lemma 2.2 and Lemma 3.2 from the main paper. In Section D.1, we prove the results controlling the number of outliers uniformly along all directions $v \in \mathcal{S}^{d-1}$. We then generalize these results to projections along PSD matrices in Section D.2.

### D.1 Linear Projections

We state Lemma 1 from Lugosi and Mendelson [LM19b]. We will use this result for distributions with bounded covariance.

**Lemma D.1.** *([LM19b, Lemma 1]) Let $x_1, \ldots, x_n$ be $n$ i.i.d. points from a distribution with mean zero and covariance $\Sigma \preceq I$. Let $Q_2$ be defined as follows:*

$$Q_2 = \frac{256}{\epsilon} \sqrt{\frac{\text{tr}(\Sigma)}{n}} + \frac{16}{\sqrt{\epsilon}}.$$

*Then, for a constant $c > 0$, with probability at least $1 - \exp(-c\epsilon n)$,*

$$\sup_{v \in \mathcal{S}^{d-1}} \left| \left\{ i : |v^T x_i| \geq Q_2 \right\} \right| \leq 0.25\epsilon n.$$

We state the following straightforward generalization of Lemma D.1 for distributions with bounded central moments. We give the proof for completeness.

**Lemma D.2.** *Let $x_1, \ldots, x_n$ be $n$ i.i.d. points from a distribution with mean zero and covariance $\Sigma \preceq I$. Further assume that for all $v \in \mathcal{S}^{d-1}$:*

$$(\mathbb{E}(v^T X)^k)^{\frac{1}{k}} \leq \sigma_k. \tag{13}$$

*Let $Q_k$ be defined as follows:*

$$Q_k = \Theta \left( \frac{1}{\epsilon} \sqrt{\frac{\text{tr}(\Sigma)}{n}} + \sigma_k \epsilon^{-\frac{1}{k}} \right).$$

*Then, there exists a $c > 0$, such that with probability at least $1 - \exp(-cn\epsilon)$,*

$$\sup_{v \in \mathcal{S}^{d-1}} \left| \left\{ i : |x_i^T v| \geq Q_k \right\} \right| = O(n\epsilon). \tag{14}$$

*Proof.* We follow the same strategy as in Lugosi and Mendelson [LM19b]. We first set $Q_k$ as follows:

$$Q_k = C \left( \frac{1}{\epsilon} \sqrt{\frac{\text{tr}(\Sigma)}{n}} + \sigma_k \epsilon^{-\frac{1}{k}} \right),$$

for a large enough constant $C$ to be determined later. Consider the function $\chi : \mathbb{R} \to \mathbb{R}$ defined by

$$\chi(x) = \begin{cases} 0, & \text{if } x \leq \frac{Q_k}{2}, \\ \frac{2x}{Q_k} - 1, & \text{if } x \in \left[ \frac{Q_k}{2}, Q_k \right], \\ 1, & \text{if } x \geq Q_k. \end{cases} \tag{15}$$

Therefore, $\mathbb{I}_{x^T v \geq Q_k} \leq \chi(x_i^T v) \leq \mathbb{I}_{x^T v \geq Q_k/2}$ and note that $\chi(\cdot)$ is a $\frac{2}{Q_k}$ Lipschitz. We first bound the number of points violating the upper tail bounds. The random quantity of interest is the following:

$$Z = \sup_{v \in \mathcal{S}^{d-1}} \sum_{i=1}^{n} \mathbb{I}_{x_i^T v \geq Q_k} . \tag{16}$$

We first calculate its expectation using the symmetrization principle [LT91, BLM13]. We have that

$$\mathbb{E}Z = \mathbb{E} \sup_{v \in \mathcal{S}^{d-1}} \sum_{i=1}^{n} \mathbb{I}_{x_i^T v \geq Q_k}$$

$$\leq \mathbb{E} \sup_{v \in \mathcal{S}^{d-1}} \sum_{i=1}^{n} \chi(x_i^T v)$$

$$\leq \mathbb{E} \sup_{v \in \mathcal{S}^{d-1}} \sum_{i=1}^{n} (\chi(x_i^T v) - \mathbb{E}\chi(x_i^T v)) + \sup_{v \in \mathcal{S}^{d-1}} \mathbb{E} \sum_{i=1}^{n} \chi(x_i^T v)$$

$$\leq 2\mathbb{E} \sup_{v \in \mathcal{S}^{d-1}} \sum_{i=1}^{n} \epsilon_i \chi(x_i^T v) + \sup_{v \in \mathcal{S}^{d-1}} \mathbb{E} \sum_{i=1}^{n} \chi(x_i^T v). \tag{17}$$

We bound the second term in Eq. (17) by

$$\mathbb{E}\sum_{i=1}^{n}\chi(x_i^T v) \leq \mathbb{E}\sum_{i=1}^{n}\mathbb{I}_{x_i^T v| \geq Q_k/2} = n\mathbb{P}(x_i^T v \geq Q_k/2) \leq n\mathbb{P}(x_i^T v \geq C\sigma_k \epsilon^{-\frac{1}{k}}) = O(n\epsilon),$$

by applying Markov inequality and choosing a large enough constant $C$ for $Q_k$. For the first term in Eq. (17), we upper bound $\chi(\cdot)$ using contraction principle for Rademacher averages and independence of $x_i$:

$$\mathbb{E}\sup_{v\in\mathcal{S}^{d-1}}\sum_{i=1}^{n}\epsilon_i\chi(x_i^T v) \leq \frac{2}{Q_k}\mathbb{E}\sup_{v\in\mathcal{S}^{d-1}}\sum_{i=1}^{n}\epsilon_i x_i^T v = \frac{2}{Q_k}\mathbb{E}\|\sum_i \epsilon_i x_i\| \leq n\frac{2}{Q_k}\sqrt{n\,\mathrm{tr}(\Sigma)} = O(n\epsilon),$$

where we use the covariance bound on $x_i$ and a large enough constant for $Q_k \geq (C/\epsilon)\sqrt{\mathrm{tr}(\Sigma)/n}$. Therefore, we get that $\mathbb{E}Z = O(n\epsilon)$. We can bound the wimpy variance, i.e., the quantity $\sigma^2$ in Theorem C.1, by $O(\epsilon n)$. By Talagrand's concentration C.1, we get that probability $1 - \exp(-cn\epsilon)$,

$$Z = O(n\epsilon + \sqrt{n\sigma}\sqrt{cn\epsilon}\sqrt{n\gamma} + cn\epsilon) = O(n\epsilon). \tag{18}$$

$\square$

## D.2 Matrix Projections

We will now use the results from the previous section to prove Lemma 2.2 and Lemma 3.2. The proof follows the ideas from [DL19, Proposition 1].

**Lemma D.3.** *(Formal version of Claim 2.4) Suppose that the event $\mathcal{E}_1$ holds, where $\mathcal{E}_1$ is the following*

$$\mathcal{E}_1 := \left\{ \sup_{v\in\mathcal{S}^{d-1}} |\{i : |x_i^T v| \geq Q_0\}| \leq 0.25\epsilon n \right\}.$$

*Let $Q = 8Q_0$ and $\epsilon \geq 1/n$. Then the event $\mathcal{E}$ also holds, where $\mathcal{E}$ is defined as follows:*

$$\mathcal{E} := \left\{ \sup_{M\in\mathcal{M}} |\{i : x_i^T M x_i \geq Q^2\}| \leq \epsilon n \right\}.$$

*Proof.* We follow the same proof strategy as Depersin and Lecue [DL19]. We reproduce the proof here for completeness.

Suppose that $\mathcal{E}_1$ holds but the desired event $\mathcal{E}$ does not hold. Let $M$ be such that $|\{i : x_i^T M x_i \geq Q^2\}| > \epsilon n$. Let $G$ be the Gaussian vector in $\mathbb{R}^d$ independent of $x_1, \ldots, x_n$ with distribution $\mathcal{N}(0, M)$. We will work conditionally on $x_1, \ldots, x_n$ in the remaining of the proof. Let $Z$ be the following random variable

$$Z = \sum_{i=1}^{n}\mathbb{I}_{|x_i^T G|^2 \geq 25Q_0^2} .$$

We have that $x_i^T G \sim \mathcal{N}(0, x_i^T M x_i)$. For $i$ such that $x_i^T M x_i \geq Q^2$, we have that

$$\mathbb{P}(|x_i^T G|^2 > 25Q_0^2) \geq 2\mathbb{P}(g \geq \frac{5}{8}) > 0.528,$$

where $g$ is a standard Gaussian random variable. Therefore,

$$\mathbb{E}Z = \sum_{i=1}^{n}\mathbb{P}(|x_i^T G|^2 > 25Q_0^2) \geq \epsilon n(0.528) \geq 0.528.$$

Note that $Z$ a is sum of independent indicator random variables. A Chernoff bound (see, e.g., [Ver18, Section 2.3]) states that, with probability at least $1 - (\sqrt{2/e})^{\mathbb{E}Z} > 0.05$, we have that $Z \geq \frac{\mathbb{E}Z}{2} > 0.25n\epsilon$. However, by Gaussian concentration (see, e.g., [BLM13]) we have that with probability at least $0.9999$: $\|G\| \leq 5$. Taking a union bound, we get that both of the events happen simultaneously with non-zero probability. Therefore, with non-zero probability $\exists u : \|u\| \leq 5$ and

$$\sum_{i=1}^{n}\mathbb{I}_{|x_i^T u|^2 \geq 25Q_0^2} > 0.25n\epsilon.$$

That is, $\exists v : \|v\| \leq 1$, and

$$\sum_{i=1}^{n} \mathbb{I}_{|x_i^T v|^2 \geq Q_0^2} > 0.25 n\epsilon \quad \equiv \quad \sum_{i=1}^{n} \mathbb{I}_{|x_i^T v| \geq Q_0} > 0.25 n\epsilon,$$

which is a contradiction to $\mathcal{E}_1$. This completes the proof. $\qquad \square$

We are now ready to prove Lemma 2.2 and 3.2.

*Proof.* (Proof of Lemma 2.2) Without loss of generality, we can assume $\epsilon n = \Omega(1)$. The result now follows from Lemma D.1, due to Lugosi and Mendelson [LM19b, Lemma 1], and Lemma D.3. $\quad \square$

*Proof.* (Proof of Lemma 3.2) Without loss of generality, we can assume $\epsilon n = \Omega(1)$. The result now follows from Lemma D.2, which might require a change of variables, and Lemma D.3. $\qquad \square$

### D.3 Truncation with Domain Constraint

**Lemma D.4.** *Let $w \in \Delta_{n,\epsilon}$ for some $\epsilon \leq 1/2$. Suppose that the following event $\mathcal{E}$ holds:*

$$\mathcal{E} := \left\{ \sup_{M \in \mathcal{M}} |\{i : (x_i - \mu)^T M (x_i - \mu) \geq Q^2\}| \leq \epsilon n \right\}.$$

*For a unit vector $v$, let $S_v \in [n]$ be the following multiset: $S_v = \{x_i : x_i \in S, |x_i^T v| \leq Q\}$. For a unit vector $v$, let $\overline{w}^{(v)}$ be the following distribution:*

$$\tilde{w}_i^{(v)} := \min\left( w_i, \frac{\mathbb{I}\{x_i \in S_v\}}{|S_v|} \right), \qquad \overline{w}^{(v)} := \frac{\tilde{w}^{(v)}}{\|\tilde{w}^{(v)}\|_1}. \tag{19}$$

*Let $g(\cdot)$ be defined as in Eq. (6). Then, for all unit vectors $v$, $\overline{w}^{(v)} \in \Delta_{n,4\epsilon,w}$. Moreover, the following inequalities hold:*

$$\left| \sum_{i=1}^{n} \overline{w}_i^{(v)} v^T (x_i - \mu) \right| \leq 4\epsilon Q + \left| \frac{\sum_{i \in S_v} v^T (x_i - \mu)}{|S_v|} \right| \leq 5\epsilon Q + \left| \frac{\sum_{i \in S} g(v^T (x_i - \mu))}{(1-\epsilon)n} \right|.$$

*Proof.* On the event $\mathcal{E}$, we have that $|S_v| \geq (1-\epsilon)n$ for all $v \in \mathcal{S}^{d-1}$. In order to show that $\overline{w}^{(v)} \in \Delta_{n,4\epsilon,w}$, it suffices to show that for all $v$, $\overline{w}_i^{(v)} \leq w_i/(1-4\epsilon)$. By the definition of $\overline{w}_i^{(v)}$, it is sufficient to show that $\|\tilde{w}^{(v)}\|_1 \geq 1 - 4\epsilon$. Let $u_S$ and $u_{S_v}$ denote the uniform distributions on the multi-sets $S$ and $S_v$ respectively. Let $d_{\mathrm{TV}}(p,q)$ denote the total variation distance between the distributions $p$ and $q$. First note that

$$d_{\mathrm{TV}}(w, u_{S_v}) \leq d_{\mathrm{TV}}(w, u_S) + d_{\mathrm{TV}}(u_S, u_{S_v}) \leq \frac{\epsilon}{1-\epsilon} + \frac{\epsilon}{1-\epsilon} \leq \frac{2\epsilon}{1-\epsilon} \leq 4\epsilon. \tag{20}$$

We now use the alternative characterization of total variation distance (see, e.g., [Tsy08, Lemma 2.1]):

$$d_{\mathrm{TV}}(p,q) = (1/2) \sum_{i=1}^{n} |p_i - q_i| = 1 - \sum_{i=1}^{n} \min(p_i, q_i).$$

Observe that $\tilde{w}^{(v)} = \min(w, u_{S_v})$; combining this observation with Eq. (20), we get the following lower bound on $\|\tilde{w}^{(v)}\|_1$:

$$\|\tilde{w}^{(v)}\|_1 = 1 - d_{\mathrm{TV}}(w, u_{S_v}) \geq 1 - 4\epsilon.$$

This concludes that $\overline{w}^{(v)} \in \Delta_{n,4\epsilon,w}$. We now focus our attention on the second result in the theorem statement. The first inequality follows from the fact that both distributions $\overline{w}^{(v)}$ and $u_{S_v}$ have total variation distance less than $4\epsilon$, and supported on $[-Q, Q]$. The second inequality follows from the fact that (i) $|S_v| \geq (1-\epsilon)n$, (ii) $g(\cdot)$ is identity on $S_v$, and bounded by $Q$ outside $[-Q, Q]$, and (iii) at most $\epsilon$-fraction of the points are outside $S_v$. This completes the proof. $\qquad \square$

# E  Stability for Distributions with Bounded Covariance

**Organization.** Section E.1 contains the proof of the sufficient conditions for stability under bounded covariance assumption (Claim 2.1). Section E.2 contains the arguments for deterministic rounding. Sections E.3 and E.4 contain the detailed proofs of the concentration results ommitted from the main text. We then combine the results from these sections to give the proof of Theorem 1.4 in Section E.5.

## E.1  Sufficient Conditions for Stability

The following claim simplifies the stability condition for the bounded covariance case.

**Claim E.1.** *(Claim 2.1) Let $S$ be a set such that $\|\mu_S - \mu\| \leq \sigma\delta$, and $\|\overline{\Sigma}_S - \sigma^2 I\| \leq \sigma^2\delta^2/\epsilon$ for some $0 \leq \epsilon \leq \delta$. Let $\epsilon' < 0.5$. Then $S$ is $(\epsilon', \delta')$ stable with respect to $\mu$ and $\sigma^2$, where $\delta' = 2\sqrt{\epsilon'} + 2\delta\sqrt{\epsilon'/\epsilon}$.*

*Proof.* Let $\epsilon' < 0.5$. Without loss of generality, we can assume that $\sigma = 1$. For $S' \subseteq S : |S'| \geq (1 - \epsilon')|S|$,

$$\frac{1}{|S'|}\sum_{i \in S'}(x_i^T v)^2 - 1 \leq \frac{1}{|S'|}\sum_{i \in S}(x_i^T v)^2 - 1 \leq \frac{1}{1-\epsilon'}(1 + \frac{\delta^2}{\epsilon}) - 1$$

$$= \frac{\frac{\delta^2}{\epsilon} + \epsilon'}{1 - \epsilon'} \leq \frac{1}{\epsilon'}(2\epsilon' + 2\delta\sqrt{\frac{\epsilon'}{\epsilon}})^2 \leq \frac{(\delta')^2}{\epsilon'}.$$

As $\delta' \geq \sqrt{\epsilon'}$, the lower bound on eigenvalues of $\overline{\Sigma}_{S'}$ is trivially satisfied. We now bound the deviation in mean. Observe that the uniform distribution on $S'$ can be obtained by conditioning the uniform distribution on $S$ on an event $E$, such that $\mathbb{P}(E) \geq 1 - \epsilon'$. Using this observation in conjunction with Hölder's inequality gives us that for any $v$, the shift in mean is at most

$$\left|\frac{1}{|S'|}\sum_{i \in S'}v^T x_i - \frac{1}{|S|}\sum_{i \in S'}v^T x_i\right| \leq 2\sqrt{1 + \frac{\delta^2}{\epsilon}}\sqrt{\epsilon'} \leq 2\sqrt{\epsilon'} + 2\delta\sqrt{\frac{\epsilon'}{\epsilon}} \leq \delta'. \tag{21}$$

$\square$

## E.2  Deterministic Rounding of the Weight Function

The next lemma states that it suffices to find a distribution $w \in \Delta_{n,\epsilon}$ for stability.

**Lemma E.2.** *For $\epsilon \leq \frac{1}{3}$, let $w^* \in \Delta_{n,\epsilon}$ be such that for $\epsilon \leq \delta$, we have*

1. $\|\mu_w - \mu\| \leq \sigma\delta$.

2. $\|\overline{\Sigma}_w - \sigma^2 I\| \leq \sigma^2\delta^2/\epsilon$.

*Then there exists a subset $S_1 \subseteq S$ such that*

1. $|S_1| \geq (1 - 2\epsilon)|S|$.

2. $S_1$ *is* $(\epsilon', \delta')$ *stable with respect to $\mu$ and $\sigma^2$, where $\delta' = O(\delta + \sqrt{\epsilon} + \sqrt{\epsilon'})$.*

*Proof.* Without loss of generality, we will assume that $\sigma^2 = 1$. We will use Claim E.1 to prove this result by first showing that there exists a $S' \subseteq [n]$ with bounded covariance and good sample mean.

Without loss of generality, we will assume that $\epsilon n$ is an integer and $\mu = 0$. We will also assume that $\frac{1}{(1-\epsilon)n} \geq w_1 \geq w_2 \geq \cdots \geq w_n \geq 0$. For any $k \in [n]$, we have that

$$1 = \sum_i w_i \leq \frac{n-k}{(1-\epsilon)n} + kw_k \tag{22}$$

$$\implies w_k \geq \frac{1}{k}\frac{(1-\epsilon)n - (n-k)}{(1-\epsilon)n} = \frac{k - \epsilon n}{(1-\epsilon)nk}. \tag{23}$$

Setting $k = 2\epsilon n$, we have that

$$w_k \geq \frac{2\epsilon n}{2n(1-\epsilon)} = \frac{1}{2(1-\epsilon)n}. \tag{24}$$

We now have a lower bound on $w_i$ for all $i \leq (1-2\epsilon)n$. Now let $S_1$ be the set of the $n-k$ points with the largest $w_i$. In particular, for each $i \in S_1$, $w_i \geq \frac{1}{2(1-\epsilon)n}$. We have that,

$$
\begin{aligned}
\sum_{i \in S_1} \frac{1}{|S_1|}(x_i^T v)^2 &= \sum_{i \in S_1} \frac{1}{(1-2\epsilon)n}(x_i^T v)^2 \\
&\leq \sum_{i \in S_1} \frac{1}{(1-2\epsilon)} 2w_i(1-\epsilon)(x_i^T v)^2 && \text{(Using Eq. (24))} \\
&\leq \frac{2(1-\epsilon)}{(1-2\epsilon)} \sum_{i \in S} w_i(x_i^T v)^2 \\
&\leq 9(1+\frac{\delta^2}{\epsilon}). \tag{25}
\end{aligned}
$$

Let the uniform distribution on $S_1$ be $u^{(1)}$ and the uniform distribution on $S$ be $u$. We now calculate the total variation distance between $w$ and $u^{(1)}$.

$$d_{\text{TV}}(w, u^{(1)}) \leq d_{\text{TV}}(w, u) + d_{\text{TV}}(u, u^{(1)}) \leq \epsilon + 2\epsilon = 3\epsilon. \tag{26}$$

Therefore, there exist distributions $p^{(1)}, p^{(2)}, p^{(3)}$ such that

$$w = (1-3\epsilon)p^{(1)} + 3\epsilon p^{(2)}, \qquad u_1 = (1-3\epsilon)p^{(1)} + 3\epsilon p^{(3)}.$$

This decomposition follows from an alternate characterization of total variation distance(see, e.g., [Tsy08, Lemma 2.1]). We first note that

$$3\epsilon \sum_i p_i^{(2)}(x_i^T v)^2 \leq \sum_i w_i(x_i^T v)^2 \leq 1 + \frac{\delta^2}{\epsilon}.3\epsilon \sum_i p_i^{(3)}(x_i^T v)^2 \leq \sum_i u_i^{(1)}(x_i^T v)^2 \leq 9\left(1 + \frac{\delta^2}{\epsilon}\right).$$

Therefore, we get that

$$
\begin{aligned}
|\sum_{i=1}^n (1-3\epsilon)p_i^{(1)} x_i^T v| &\leq |\sum_{i=1}^n w_i x_i^T v| + |3\epsilon \sum_i p_i^{(3)} x_i^T v| \leq \delta + 3\epsilon \sqrt{\sum_{i=1}^n p_i(x_i^v)^2} \\
&\leq \delta + \sqrt{3\epsilon} \sqrt{3\epsilon \sum_{i=1}^n p_i(x_i^T v)^2} \leq \delta + \sqrt{3\epsilon} \sqrt{(1 + \frac{\delta^2}{\epsilon})} \\
&\leq \delta + \sqrt{3\epsilon} + \sqrt{3}\delta \leq 3\delta + 2\sqrt{\epsilon}.
\end{aligned}
$$

We finally get that

$$
\begin{aligned}
|\sum_{i=1}^n u_i^{(1)} x_i^T v| &\leq |\sum_{i=1}^n (1-3\epsilon)p_i^{(1)} x_i^T v| + |\sum_{i=1}^n 3\epsilon p_i^{(3)} x_i^T v| \\
&\leq 3\delta + 2\sqrt{\epsilon} + \sqrt{3\epsilon} \sqrt{3\epsilon \sum_i p_i^{(3)}(x_i^T v)^2} \\
&\leq 3\delta + 2\sqrt{\epsilon} + \sqrt{27}\sqrt{\epsilon + \delta^2} \leq 10\delta + 10\sqrt{\epsilon}. \tag{27}
\end{aligned}
$$

Therefore using Equations (25) and (27), we have a set $S_1$ that satisfies the conditions in Claim E.1 with $\delta'' = 10\delta + 10\sqrt{\epsilon}$. Using Claim E.1, we get that $S_1$ is $(\epsilon', \delta')$ stable. $\qquad \square$

### E.3 Controlling the Variance

We provide the details of concentration of the empirical process, related to the variance in Lemma 2.3, which was ommitted from the main text.

**Lemma E.3.** *Consider the setting in the proof of Lemma 2.3. Then, with probability $1 - \exp(-n\epsilon)$, $R'/n \leq \delta^2/\epsilon$, where $\delta = O(\sqrt{(\text{tr}(\Sigma)\log \text{r}(\Sigma))/n} + \sqrt{\epsilon})$.*

*Proof.* We will apply Talagrand's concentration inequality for the bounded empirical process, see Theorem C.1. We first calculate the quantity $\sigma^2$, the wimpy variance, required in Theorem C.1 below

$$\sigma^2 = \sup_{M \in M} \sum_{i=1}^n \mathbb{V}(f(x_i^T M x_i)) \leq \sup_{M \in M} \sum_{i=1}^n \mathbb{E}(f(x_i^T M x_i))^2 \leq \sup_{M \in M} \sum_{i=1}^n Q^2 \mathbb{E} f(x_i^T M x_i) \leq nQ^2,$$

where we use that $f(x) \leq Q^2$, $f(x) \leq x$, and $\mathbb{E} x^T M x \leq 1$. We now focus our attention to $\mathbb{E} R'$. Let $\xi_i$ be $n$ i.i.d. Rademacher random variables, independent of $x_1, \ldots, x_n$. We use contraction and symmetrization properties for Rademacher averages [LT91, BLM13] to get

$$\mathbb{E} R' = \mathbb{E} \sup_{M \in \mathcal{M}} \sum_{i=1}^n f(x_i^T M x_i) - \mathbb{E} f(x_i^T M x_i) \leq 2\mathbb{E} \left| \sup_{M \in \mathcal{M}} \sum_{i=1}^n \xi_i f(x_i^T M x_i) \right|$$

$$\leq 2\mathbb{E} \left| \sup_{M \in \mathcal{M}} \sum_{i=1}^n \xi_i x_i^T M x_i \right| \leq 2\mathbb{E} \sup_{M \in \mathcal{M}} \| \sum_{i=1}^n \xi_i x_i x_i^T \| \| \text{tr}(M) \| \tag{28}$$

$$= 2\mathbb{E} \| \sum_{i=1}^n \xi_i x_i x_i^T \| = O\left( \sqrt{\frac{n\, \text{tr}(\Sigma)\log \text{r}(\Sigma)}{\epsilon}} + \frac{\text{tr}(\Sigma)\log \text{r}(\Sigma)}{\epsilon} \right),$$

where the last step uses the refined version of matrix-Bernstein inequality [Min17], stated in Theorem C.2, with $L = O(\text{tr}(\Sigma)/\epsilon)$.

Note that the empirical process $R'$ is bounded by $Q^2$. By applying Talagrand's concentration inequality for bounded empirical processes (Theorem C.1), with probability at least $1 - \exp(-n\epsilon)$, we have

$$R' = O\left( \mathbb{E} R' + \sqrt{nQ^2}\sqrt{n\epsilon} + Q^2 n\epsilon \right)$$

$$\implies \frac{R'}{n} = O\left( \frac{\text{tr}(\Sigma)\log \text{r}(\Sigma)}{n\epsilon} + \sqrt{\frac{\text{tr}(\Sigma)\log \text{r}(\Sigma)}{n\epsilon}} + Q\sqrt{\epsilon} + \epsilon Q^2 \right)$$

$$= \frac{1}{\epsilon} O\left( \frac{\text{tr}(\Sigma)\log \text{r}(\Sigma)}{n} + \sqrt{\frac{\text{tr}(\Sigma)\log \text{r}(\Sigma)}{n}}\sqrt{\epsilon} + Q\epsilon\sqrt{\epsilon} + (\epsilon Q)^2 \right)$$

$$= \frac{1}{\epsilon} \left( O\left( \sqrt{\frac{\text{tr}(\Sigma)\log \text{r}(\Sigma)}{n}} + \sqrt{\epsilon} + \epsilon Q \right) \right)^2$$

$$= \frac{\delta^2}{\epsilon},$$

where $\delta = O(\sqrt{\text{tr}(\Sigma)\log \text{r}(\Sigma)/n} + \sqrt{\epsilon} + \epsilon Q) = O(\sqrt{\text{tr}(\Sigma)\log \text{r}(\Sigma)/n} + \sqrt{\epsilon})$, where we use the fact that $\epsilon Q = O(\sqrt{\epsilon} + \sqrt{\text{tr}(\Sigma)/n})$. $\qquad\square$

### E.4 Controlling the Mean

We now provide the detailed argument for the concentration of empirical process related to mean in Lemma 2.5, which was ommitted from the main text.

**Lemma E.4.** *Consider the setting in Lemma 2.5. Then, with probability, $1 - \exp(-n\epsilon)$, $R'/n = O(\sqrt{\text{tr}(\Sigma)/n} + \sqrt{\epsilon})$.*

*Proof.* We will use Talagrand's concentration inequality for bounded empirical processes, stated in Theorem C.1. We first calculate the wimpy variance required for Theorem C.1,

$$\sigma^2 = \sup_{v \in \mathcal{S}^{d-1}} \sum_{i=1}^n \mathbb{V}(g(x_i^T v)) \leq \sup_{v \in \mathcal{S}^{d-1}} \sum_{i=1}^n \mathbb{E} g(v^T x_i)^2 \leq \sup_{v \in \mathcal{S}^{d-1}} n\mathbb{E}(v^T x_i)^2 \leq n. \tag{29}$$

We also bound the quantity $\mathbb{E}R'$ using symmetrization and contraction [LT91, BLM13] properties of Rademacher averages. We have that

$$\mathbb{E}R' = \mathbb{E}\sup_{v \in \mathcal{S}^{d-1}} \sum_{i=1}^{n} g(v^T x_i) - \mathbb{E}g(v^T x_i) \leq 2\mathbb{E}\left|\sup_{v \in \mathcal{S}^{d-1}} \sum_{i=1}^{n} \epsilon_i g(v^T x_i)\right|$$

$$\leq 2\mathbb{E}\left|\sup_{v \in \mathcal{S}^{d-1}} \sum_{i=1}^{n} \epsilon_i v^T x_i\right| = 2\mathbb{E}\|\sum_{i=1}^{n} \epsilon_i x_i\| \leq 2\sqrt{n\operatorname{tr}(\Sigma)},$$

where the last step uses that $\epsilon_i x_i$ has covariance $\Sigma$. By applying Talagrand's concentration inequality for bounded empirical processes (Theorem C.1), we get that with probability at least $1 - \exp(-n\epsilon)$,

$$R'/n = O(\mathbb{E}R'/n + \sqrt{n\epsilon} + Q\epsilon) = O(\sqrt{\operatorname{tr}(\Sigma)/n} + \sqrt{\epsilon}).$$

$\square$

## E.5 Proof of the Main statement

**Theorem E.5.** *(Theorem 1.4) Let $x_1, \ldots, x_n$ be $n$ i.i.d. points in $\mathbb{R}^d$ from a distribution with mean $\mu$ and covariance $\Sigma$. Let $\epsilon' = \Theta(\log(1/\tau)/n + \epsilon) \leq c$ for a sufficiently small positive constant $c$. Then, with probability at least $1 - \tau$, there exists a subset $S' \subseteq S$ s.t. $|S|' \geq (1 - \epsilon')n$ and $|S'|$ is $(C\epsilon', \delta)$-stable with respect to $\mu$ and $\|\Sigma\|$ with $\delta = O(\sqrt{(\operatorname{r}(\Sigma)\log\operatorname{r}(\Sigma))/n} + \sqrt{C\epsilon'})$.*

*Proof.* Note that we can assume without loss of generality that $\mu = 0$ and $\|\Sigma\| = 1$, upper bound $\delta$ by $\delta = O(\sqrt{\operatorname{tr}(\Sigma)\log(\operatorname{r}(\Sigma))/n} + \sqrt{C\epsilon'})$; otherwise, apply the following arguments to the random variable $(x_i - \mu)/\sqrt{\|\Sigma\|}$ (the result holds trivially if $\|\Sigma\| = 0$).

We first prove a simpler version of the theorem for distributions with bounded support. The reason we make this assumption is to apply the matrix concentration results in Theorem C.2.

**Base case: Bounded support** Assume that $\|x_i - \mu\| = O(\sqrt{\operatorname{tr}(\Sigma)/\epsilon'})$ almost surely.

Note that the bounded support assumption allows us to apply Lemma 2.3. Set $\tilde{\epsilon} = \epsilon'/c'$ for a large constant $c'$ to be determined later. Let $u^* \in \Delta_{n,\tilde{\epsilon}}$ achieve the minimum in Lemma 2.3. For this $u^*$, let $w^* \in \Delta_{n,4\tilde{\epsilon},u^*}$ be the distribution achieving the minimum in Lemma 2.5. Note that the probability of error is at most $2\exp(-\Omega(n\tilde{\epsilon}))$. We can choose $\epsilon'$ large enough, $\tilde{\epsilon} = \epsilon'/c = \Omega(\log(1/\tau)/n)$, so that the probability of failure is at most $1 - \tau$. Let $\delta = C\sqrt{\operatorname{tr}(\Sigma)\log\operatorname{r}(\Sigma)/n} + C\sqrt{\tilde{\epsilon}}$ for a large enough constant $C$ to be determined later. We first look at the variance of $w^*$ using the guarantee of $u^*$ in Lemma 2.3:

$$\sum_{i=1}^{n} w_i^* x_i x_i^T \preceq \sum_{i=1}^{n} \frac{1}{1-\epsilon'} u_i^* x_i x_i^T \preceq 2\sum_{i=1}^{n} u_i^* x_i x_i^T \leq \frac{1}{\tilde{\epsilon}}(C\sqrt{\operatorname{tr}(\Sigma)\log\operatorname{r}(\Sigma)/n} + C\sqrt{\tilde{\epsilon}})^2. \quad (30)$$

By choosing $C$ to be a large enough constant, we get that $\|\sum_{i=1}^{n} w^* x_i x_i^T - I\| \leq \delta^2/\tilde{\epsilon}$. Now, we look at the mean. Lemma 2.5 states that

$$\left\|\sum_{i=1}^{n} w^* x_i\right\| = O\left(\sqrt{\tilde{\epsilon}} + C\sqrt{\frac{\operatorname{tr}(\Sigma)}{n}}\right) \leq \delta. \quad (31)$$

Since $w^* \in \Delta_{n,4\tilde{\epsilon},u^*}$ and $u^* \in \Delta_{n,\tilde{\epsilon}}$, we have that $w^* \in \Delta_{n,5\tilde{\epsilon}}$. Therefore, we have a $w^* \in \Delta_{n,5\tilde{\epsilon}}$ that satisfies the requirements of Lemma E.2. Applying Lemma E.2, we get the desired statement for a set $S' \subseteq S$. Finally, we can choose the constant $c'$ in the definition of $\tilde{\epsilon}$ large enough, so that the set has cardinality $|S'| \geq (1 - \epsilon')n$. This completes the proof for the case of bounded support.

**General case** We first do a simple truncation. For a large enough constant $C'$, let $E$ be the following event:

$$E = \left\{X : \|X - \mu\| \leq C'\sqrt{\frac{\operatorname{tr}(\Sigma)}{\epsilon'}}\right\}. \quad (32)$$

Let $Q$ be the distribution of $X$ conditioned on $E$. Note that $P$ can be written as a convex combination of two distributions: $Q$ and some distribution $R$,

$$P = (1 - \mathbb{P}(E))Q + \mathbb{P}(E^c)R. \tag{33}$$

Let $Z \sim Q$. By Chebyshev's inequality, we get that $\mathbb{P}(E^c) \leq \epsilon'/C'^2$. Using Lemma C.5, we get that $\|\mathbb{E}Z - \mu\| = O(\sqrt{\epsilon'})$ and $\mathrm{Cov}(Z) \preceq I$. The distribution $Q$ satisfies the assumptions of the base case analyzed above. Let $S_E$ be the set $\{i : x_i \in E\}$ and let $E_1$ be the following event:

$$E_1 = \{|S_E| \geq (1 - \epsilon'/2)n\}. \tag{34}$$

A Chernoff bound implies that given $n$ samples from $P$, for a $c > 0$, with probability at least $1 - \exp(-cn\epsilon'/C'^2) \geq 1 - \tau/2$ (by choosing $C'$ large enough and $\epsilon' = \Omega(\log(1/\tau)/n))$, $E_1$ holds.

For a fixed $m \geq (1 - \epsilon'/2)n$, let $z_1, \ldots, z_m$ be $m$ i.i.d. draws from the distribution $Q$. Applying the theorem statement of the base case for each such $m$, we get that, except with probability $\tau/2$, there exists an $S' \subseteq [m] \subseteq [n]$ with $|S'| \geq (1 - \epsilon'/2)m \geq (1 - \epsilon'/2)^2 n \geq (1 - \epsilon')n$, such that $|S'|$ is $(C\epsilon', O(\sqrt{d \log d/n} + \sqrt{C\epsilon'}))$-stable.

As mentioned above (event $E_1$), $m \geq (1 - \epsilon'/2)n$ with probability at least $1 - \tau/2$. We can now marginalize over $m$ to say that with probability at least $1 - \tau$, there exists a $(C\epsilon', \delta)$ stable set $S'$ of cardinality at least $(1 - \epsilon')n$.

However, we are still not done. We have the guarantee that $S'$ is stable with respect to $\mathbb{E}Z$. Using the triangle inequality and Cauchy-Schwarz, we get that the set is also $(C\epsilon', \delta')$ stable with respect to $\mu$ as well, where $\delta' = \delta + \|\mu - \mathbb{E}Z\| = \delta + O(\sqrt{\epsilon'})$. This completes the proof. □

# F    Stability for Distributions with Bounded Central Moments

**Organization.**   In this section, we give the detailed arguments for the proof of Theorem 1.8. Our proof strategy closely follows the proof structure of the bounded covariance case. We suggest the reader to read Section 2 before reading this section.

This section has a similar organization to Appendix E. We start with a simplified stability condition in Section F.1. Section F.2 contains the argument for rounding a good distribution $w \in \Delta_{n,\epsilon}$ to a subset. Sections F.3 and F.4 contain the arguments for controlling the second moment matrix from above and below respectively. Section F.5 contains the results regarding the concentration results for controlling the sample mean. Finally, we combine the results of the previous sections in Section F.6 to complete the proof of Theorem 1.8.

## F.1    Sufficient Conditions for Stability

We will prove the existence of a stable set with high probability using the following claim. This is analogous to Claim E.1 in the bounded covariance setting, but we also need a lower bound on the minimum eigenvalue of $\overline{\Sigma}_{S'}$ for all large subsets $S'$.

**Claim F.1.** *Let $0 \leq \epsilon \leq \delta$ and $\epsilon \leq 0.5$. A set $S$ is $(\epsilon, 7\delta)$ stable, if it satisfies the following for all unit vectors $v$.*

1. *$\|\mu_S - \mu\| \leq \delta$.*

2. *$v^T \overline{\Sigma}_S v \leq 1 + \frac{\delta^2}{\epsilon}$.*

3. *For all subsets $S' \subseteq S$ such that $|S'| \geq (1 - \epsilon)|S|$, we have $v^T \overline{\Sigma}_{S'} v \geq (1 - \frac{\delta^2}{\epsilon})$.*

*Proof.* Without loss of generality, we will assume that $\mu = 0$. We first show the second condition in the definition of stability. Let $S'$ be any proper subset of $S$, such that $|S'| \geq (1 - \epsilon)|S|$. Note that the minimum eigenvalue of $S'$ is lower-bounded by the assumption:

$$v^T \Sigma_{S'} v = \frac{1}{|S \setminus S_\epsilon|} \sum_{i \in S \setminus S_\epsilon} (v^T x)^2 \geq 1 - \frac{\delta^2}{\epsilon}. \tag{35}$$

We now look at the largest eigenvalue of $S'$:

$$v^T \Sigma_S v - 1 = \frac{1}{|S'|} \sum_{i \in S'} (v^T x)^2 - 1 \leq \frac{|S|}{|S'|} \frac{1}{|S|} \sum_{i \in S} (v^T x)^2 - 1$$

$$\leq \frac{1}{1 - \epsilon} (1 + \frac{\delta^2}{\epsilon}) - 1 \leq \frac{1}{1 - \epsilon} (\frac{\delta^2}{\epsilon} + \epsilon) \leq \frac{2\delta^2}{\epsilon} + 2\epsilon \leq 4 \frac{\delta^2}{\epsilon}.$$

We now need to show that the mean of $S'$ is also good. In order to do that, we first control the deviation due to a small set $S \setminus S'$.

$$\frac{1}{|S|} \sum_{i \in S \setminus S'} (v^T x_i)^2 = \frac{1}{|S|} \sum_{i \in S} (v^T x_i)^2 - \frac{1}{|S|} (\sum_{i \in S'} (v^T x_i)^2)$$

$$\leq (1 + \frac{\delta^2}{\epsilon}) - \frac{|S'|}{|S|} (1 - \frac{\delta^2}{\epsilon})$$

$$\leq (1 + \frac{\delta^2}{\epsilon}) - (1 - \epsilon)(1 - \frac{\delta^2}{\epsilon}) \leq \frac{2\delta^2}{\epsilon} + \epsilon. \tag{36}$$

We first break the deviation in mean into two terms, and control each individually:

$$\left| \frac{1}{|S'|} \sum_{i \in S'} (v^T x_i) \right| = \frac{|S|}{|S'|} \left| \frac{1}{|S|} \sum_{i \in S \setminus S_\epsilon} (v^T x_i) \right| \leq \frac{|S|}{|S'|} \left| \frac{1}{|S|} \sum_{i \in S} (v^T x_i) \right| + \frac{|S|}{|S'|} \left| \frac{1}{|S|} \sum_{i \in S \setminus S'} (v^T x_i) \right|.$$

We can upper bound the first term by $\|\mu_S\|/(1 - \epsilon) \leq \delta/(1 - \epsilon)$. We bound the second term using the Cauchy-Schwarz inequality and Eq. (36):

$$\frac{|S|}{|S'|} \left| \frac{1}{|S|} \sum_{i \in S \setminus S'} (v^T x_i) \right| \leq \frac{|S \setminus S'|}{|S'|} \left| \frac{1}{|S \setminus S'|} \sum_{i \in S \setminus S'} (v^T x_i) \right|$$

$$\leq \frac{|S \setminus S'|}{|S'|} \sqrt{\frac{1}{|S \setminus S'|} \sum_{i \in S \setminus S'} (v^T x_i)^2}$$

$$= \frac{\sqrt{|S \setminus S'||S|}}{|S'|} \sqrt{\frac{1}{|S|} \sum_{i \in S \setminus S'} (v^T x_i)^2} \leq \frac{\sqrt{\epsilon}}{1 - \epsilon} \sqrt{\frac{2\delta^2}{\epsilon} + \epsilon}.$$

Overall, we get that

$$|v^T \mu_{S'}| \leq \frac{1}{1 - \epsilon} (\delta + \sqrt{2}\delta + \epsilon) \leq 5\delta + 2\epsilon \leq 7\delta.$$

$\square$

## F.2  Randomized Rounding of Weight Function

In this section, we show how to recover a subset from a $w \in \Delta_{n,\epsilon}$. Unlike the deterministic rounding in Section E.2, we do a randomized rounding in Lemma F.2 to get a better dependence on $\epsilon$. For the second condition ($\delta^2 = O(\epsilon)$) in Lemma F.2 to hold, it is necessary that $n = \Omega(d)$. If $n = O(d)$, it is not a problem because, in this regime, the bounded covariance assumption already leads to optimal error.

**Lemma F.2.** *Let $k \geq 4$. Let $w \in \Delta_{n,\epsilon}$, for $\epsilon \leq \frac{1}{3}$, be a distribution on the set of points $S$ such that*

1. $\|\mu_w - \mu\| \leq \delta$.

2. $\|\overline{\Sigma}_w\| - 1 \leq \frac{\delta^2}{\epsilon} \leq r_1$, *for some $r_1 > 1$.*

3. *Let $C \geq 4$. For all subsets $S'$: $|S'| \geq (1 - C\epsilon)n$ and $v \in \mathcal{S}^{d-1}$: $v^T \overline{\Sigma}_{S'} v \geq 1 - \delta^2/(C\epsilon)$.*

4. $w_i > 0$ *implies that $\|x_i\| \leq r_2 \sigma_k \sqrt{d} \gamma^{-1/k}$ for some $r_2 \geq 1$.*

*Then, there exists a subset $S_1 \subseteq [n]$ such that*

1. $|S_1| \geq (1 - 2\epsilon)n.$

2. $S_1$ *is* $(\epsilon', \delta')$ *stable, where*

$$\epsilon' = (C - 2)\epsilon, \qquad \delta' = O\left(\delta + \sqrt{\frac{r_1 d \log d}{n}} + r_2 \sigma_k \epsilon^{\frac{1}{2} - \frac{1}{k}}\sqrt{\frac{d \log d}{n}} + r_2 \sigma_k \epsilon^{1 - \frac{1}{k}}\right). \tag{37}$$

*Proof.* We will use Claim F.1 to prove this result. Without loss of generality, let $\mu = 0$. Therefore, it suffices to find a subset such that both the mean and the largest eigenvalue are controlled. Let $Y_i \sim \text{Bernoulli}(w_i(1 - \epsilon)n)$. We have that $\sum_{i=1}^n \mathbb{E} Y_i = (1 - \epsilon)n$. Let $S_1$ be the (random) set:

$$S_1 = \{i : Y_i = 1\}. \tag{38}$$

By a Chernoff bound, we have that for some constant $c' > 0$,

$$\mathbb{P}(|S_1| \geq (1 - 2\epsilon)n) \leq \exp(-c' n \epsilon). \tag{39}$$

Let $E$ be the event $E = \{|S_1| \geq (1 - 2\epsilon)n\}$. We now bound the mean of the set $S_1$. Consider the following random variable $Z$:

$$Z = \sum_i (Y_i - (1 - \epsilon)w_i n)x_i. \tag{40}$$

The random variable $Z$ satisfies $\mathbb{E} Z = 0$. Moreover, its covariance can be bounded using the assumption as follows:

$$v^T \Sigma_Z v = \sum_{i=1}^n w_i(1 - \epsilon)n(1 - w_i(1 - \epsilon)n)(v^T x_i)^2$$

$$\leq (1 - \epsilon)n \sum_{i=1}^n w_i(x_i^T v)^2 \leq (1 - \epsilon)n(1 + \frac{\delta^2}{\epsilon}) \preceq 2r_1 n.$$

Therefore, with probability at least $0.8$, we have that

$$\|Z\| \leq 10\sqrt{r_1 n d}$$

$$\implies \|\sum Y_i x_i\| \leq (1 - \epsilon)n\|\sum_i w_i X_i\| + 10\sqrt{r_1 n d}.$$

Let $E_2$ be the event that $E_2 = \{\|\sum Y_i x_i\| \leq (1 - \epsilon)n\sigma + 10\sqrt{r_1 n d}\}$. This implies that on the event $E \cap E_1$,

$$\|\mu_{S_1}\| \leq \frac{1 - \epsilon}{1 - 2\epsilon}\delta + 10\frac{c_5}{1 - 2\epsilon}\sqrt{\frac{d}{n}} \leq 2\delta + 30\sqrt{\frac{r_1 d}{n}}. \tag{41}$$

We now focus our attention on upper bounding the eigenvalue. Define the symmetric random matrix, $Z_i$ as $Z_i := Y_i x_i x_i^T - w_i(1 - \epsilon)n x_i x_i^T$. We have that $\mathbb{E} Z_i = 0$ and $\|Z_i\| \leq r_2^2 d \sigma_k \epsilon^{1 - \frac{1}{k}}$ almost surely. We now bound the matrix variance statistic (used in Theorem C.2):

$$\nu(Z) = \left\|\sum_{i=1}^n w_i(1 - \epsilon)n(1 - w_i(1 - \epsilon)n)\|x_i\|^2 x_i x_i^T\right\|$$

$$\leq \left\|\sum_{i=1}^n w_i(1 - \epsilon)n\frac{r_2^2 \sigma_k^2 d}{\epsilon^{\frac{2}{k}}} x_i x_i^T\right\|$$

$$\leq (1 - \epsilon)\frac{r_2^2 \sigma_k^2 n d}{\epsilon^{\frac{2}{k}}}\left\|\sum_{i=1}^n w_i x_i x_i^T\right\|$$

$$\leq (1 - \epsilon)\frac{r_2^2 \sigma_k^2 n d}{\epsilon^{\frac{2}{k}}}\|\overline{\Sigma}_w\| \leq 2\frac{r_1 r_2^2 \sigma_k^2 n d}{\epsilon^{\frac{2}{k}}}.$$

By the matrix concentration (Theorem C.2), we get that with probability at least 0.8, we have that

$$\left\| \sum_{i=1}^n Y_i x_i x_i^T - w_i(1-\epsilon)n x_i x_i^T \right\| = O\left( \sqrt{\frac{r_1 r_2^2 \sigma_k^2 n d \log d}{\epsilon^{\frac{2}{k}}}} + \frac{r_2^2 \sigma_k^2 d \log d}{\epsilon^{\frac{2}{k}}} \right). \tag{42}$$

Let $E_3$ be the event above, which happens with probability at least 0.8. Under the event $E \cap E_3$, we get that

$$\begin{aligned}
v^T \overline{\Sigma}_{S_1} v &\leq \frac{1-\epsilon}{1-2\epsilon} w_i (x_i^T v)^2 + \frac{1}{1-2\epsilon} O\left( \sqrt{\frac{r_1 r_2^2 \sigma_k^2 d \log d}{n \epsilon^{\frac{2}{k}}}} + \frac{r_2^2 \sigma_k^2 d \log d}{n \epsilon^{\frac{2}{k}}} \right) \\
&\leq \frac{1-\epsilon}{1-2\epsilon}(1 + \frac{\delta^2}{\epsilon}) + O\left( \sqrt{\frac{r_1 r_2^2 \sigma_k^2 d \log d}{n \epsilon^{\frac{2}{k}}}} + \frac{r_2^2 \sigma_k^2 d \log d}{n \epsilon^{\frac{2}{k}}} \right) \\
&\leq 1 + \frac{1}{\epsilon} O\left( \epsilon^2 + \delta^2 + \sqrt{\frac{d \log d}{n}} r_1 r_2 \sigma_k \epsilon^{1-\frac{1}{k}} + r_2^2 \sigma_k^2 \epsilon^{1-\frac{2}{k}} \frac{d \log d}{n} \right) \\
&\leq 1 + \frac{1}{\epsilon}\left( O\left( \delta + r_1 r_2 \sigma_k \epsilon^{1-\frac{1}{k}} + \sqrt{\frac{d \log d}{n}} + r_2 \sigma_k \epsilon^{\frac{1}{2}-\frac{1}{k}} \sqrt{\frac{d \log d}{n}} \right) \right)^2. \tag{43}
\end{aligned}$$

Let $\epsilon' = (C-2)\epsilon$. Note that if $|S_1| \geq (1-2\epsilon)|S|$, then $|S'| \geq (1-\epsilon')|S_1|$ implies that $|S'| \geq (1-C\epsilon)|S|$, which leads to a lower bound on the minimum eigenvalue. This follows from the following elementary calculations:

$$\frac{|S'|}{|S|} \geq (1-2\epsilon)\frac{|S_1|}{|S|} \geq (1-2\epsilon)(1-(C-2)\epsilon) \geq 1 - C\epsilon. \tag{44}$$

Using Equations (39), (41) and (43), we get that there exists a subset $S_1$ such that for all $v \in \mathcal{S}^{d-1}$ and $\delta' = O(\delta + \sqrt{r_1 d \log d/n} + r_1 r_2 \sigma_k \epsilon^{1/2-1/k} \sqrt{d \log d/n} + r_1 r_2 \sigma_k \epsilon^{1-\frac{1}{k}})$:

1. $|S_1| \geq (1-2\epsilon)n \geq (1-\epsilon')n$.

2. $\|\mu_{S_1}\| \leq \delta'$.

3. $v^T \overline{\Sigma}_{S_1} v \leq 1 + \frac{\delta'^2}{\epsilon'}$.

4. For all subsets $S' \subseteq S_1 : |S'| \geq (1-\epsilon')|S_1|, v^T \overline{\Sigma}_{S'} v \geq 1 - \frac{\delta'^2}{\epsilon'}$.

We now invoke Claim F.1 to conclude that $S'$ is $(\epsilon', 7\delta')$-stable. □

## F.3 Upper Bound on the Second Moment Matrix

As $\Sigma = \sigma^2 I$, we have that $\text{tr}(\Sigma) = \sigma^2 d$, and $\text{r}(\Sigma) = d$. We follow the same strategy as in Section 2.1. We first find a subset such that its covariance matrix is bounded. For technical reasons, we do not assume that the covariance is exactly identity and allow some slack. The argument is similar to Lemma 2.3 for the bounded covariance. We also impose some additional constraints to simplify the expression, as those regimes would not hold anyway in the proof.

**Lemma F.3.** *Let $x_1, \ldots, x_n$ be $n$ i.i.d. points in $\mathbb{R}^d$ from a distribution with mean $\mu$, covariance $\Sigma$, and for a $k \geq 4$, the $k$-th central moment is bounded by $\sigma_k$. Further assume that for $\epsilon < 0.5$, covariance matrix $\Sigma$ satisfies that $(1 - 2\sigma_k^2 \epsilon^{1-\frac{2}{k}}) \preceq \Sigma \preceq I$. Further assume the following conditions hold:*

1. $\log(1/\tau)/n = O(\epsilon)$.

2. $\|x_i - \mu\| = O(\sigma_k \sqrt{d} \epsilon^{-1/k})$ *almost surely.*

3. $\sigma_k \epsilon^{\frac{1}{2}-\frac{1}{k}} = O(1)$.

*Then, for a $c > 0$, with probability $1 - \tau - \exp(-cn\epsilon)$: $\min_{w \in \Delta_{n,\epsilon}} \|\overline{\Sigma}_w\| \leq 1 + \delta^2/\epsilon$, where $\delta = O(\sqrt{(d \log d)/n} + \sigma_k \epsilon^{1-\frac{1}{k}} + \sigma_4 \sqrt{\log(1/\tau)/n})$.*

*Proof.* We will assume without loss of generality that $\mu = 0$. We will assume that the event $\mathcal{E}$ in Lemma 3.2 holds as it only incurs an additional probability of error of $\exp(-cn\epsilon)$. We use the variational characterization of spectral norm and minimax duality to write the following:

$$\min_{w \in \Delta_{n,\epsilon}} \|\sum_i w_i x_i x_i^T\| = \min_{w \in \Delta_{n,\epsilon}} \max_{M \in \mathcal{M}} \sum w_i \langle x_i x_i^T, M \rangle$$

$$= \max_{M \in \mathcal{M}} \min_{w \in \Delta_{n,\epsilon}} \sum w_i x_i^T M x_i$$

$$\leq \max_{M \in \mathcal{M}} \sum_{i=1}^n \frac{1}{(1-\epsilon)n} (x_i^T M x_i) \mathbb{I}_{x_i^T M x_i \leq Q_k^2},$$

where the third inequality uses Lemma 3.2, where it chooses the uniform distribution on the set $S_M = \{x_i : x_i^T M x_i \leq Q_k^2\}$. Let $f : \mathbb{R}_+ \to \mathbb{R}_+$ be the following function:

$$f(x) := \begin{cases} x, & \text{if } x \leq Q_k^2 \\ Q_k^2, & \text{otherwise.} \end{cases}$$

Define the following random variables $R$ and $R'$:

$$R = \sup_{M \in \mathcal{M}} \sum_{i=1}^n f(x_i^T M x_i), \qquad R' = \sup_{M \in \mathcal{M}} \sum_{i=1}^n f(x_i^T M x_i) - \mathbb{E} f(x_i^T M x_i).$$

By Lemma C.4, we get that $|\mathbb{E} f(x_i^T M x) - 1| \leq 2\sigma_k^2 \epsilon^{1-\frac{2}{k}}$, which gives that

$$|R - n - R'| \leq 2n\sigma_k^2 \epsilon^{1-\frac{2}{k}}.$$

We therefore get that

$$\min_{w \in \Delta_{n,\epsilon}} \|\sum_i w_i x_i x_i^T\| - 1 \leq \max_{M \in \mathcal{M}} \sum_{i=1}^n \frac{1}{(1-\epsilon)n} (x_i^T M x_i) \mathbb{I}_{x_i^T M x_i \leq Q_k^2} - 1$$

$$\leq \max_{M \in \mathcal{M}} \sum_{i=1}^n \frac{1}{(1-\epsilon)n} f(x_i^T M x_i) - 1$$

$$= \frac{1}{(1-\epsilon)n} R - 1$$

$$\leq \frac{2R'}{n} + 4\sigma_k^2 \epsilon^{1-\frac{2}{k}} + 2\epsilon.$$

Observe that the last two terms in the above expression are small, i.e., $\sigma_k^2 \epsilon^{1-\frac{1}{k}} + \epsilon = O(\delta^2/\epsilon)$. We next use Lemma F.4 in Appendix to conclude that $R'$ concentrates well. Lemma F.4 states that with probability $1 - \tau$, $R'/n \leq (1/\epsilon)(O(\sqrt{d \log d/n} + \sigma_k \epsilon^{1-\frac{1}{k}} + \sigma_4 \sqrt{\log(1/\tau)/n}))^2$. Note that both of the remaining terms are small compared to Overall, we get that

$$\min_{w \in \Delta_{n,\epsilon}} \|\overline{\Sigma}_w\| \leq 1 + \frac{\delta^2}{\epsilon}.$$

Taking a union bound on the event $\mathcal{E}$ and concentration of $R'$ concludes the result. $\qquad \square$

**Lemma F.4.** *Consider the conditions in Lemma F.3. Then, with probability $1 - \tau$, $R'/n \leq \delta^2/\epsilon$, where $\delta = O(\sqrt{d \log d/n} + \sigma_k \epsilon^{1-\frac{1}{k}} + \sigma_4 \sqrt{\log(1/\tau)/n})$.*

*Proof.* (Proof of Lemma F.4) We first calculate the wimpy variance required for Theorem C.1,

$$\sigma^2 = \sup_{M \in M} \sum_{i=1}^n \mathbb{V}(f(x_i^T M x_i)) \leq \sup_{M \in M} \sum_{i=1}^n \mathbb{E} f(x_i^T M x_i)^2 \tag{45}$$

$$\leq n \sup_{M \in M} \mathbb{E}(x_i^T M x_i)^4 \leq n\sigma_4^4. \tag{46}$$

We use symmetrization, contraction, and matrix concentration (Theorem C.2) to bound $\mathbb{E}R'$ as follows:

$$\mathbb{E}R' = \mathbb{E}\sup_{M\in\mathcal{M}}\sum_{i=1}^{n} f(x_i^T M x_i) - \mathbb{E}f(x_i^T M x_i) \leq 2\mathbb{E}\left|\sup_{M\in\mathcal{M}}\sum_{i=1}^{n}\epsilon_i f(x_i^T M x_i)\right|$$

$$\leq 2\mathbb{E}\left|\sup_{M\in\mathcal{M}}\sum_{i=1}^{n}\epsilon_i x_i^T M x_i\right| \leq 2\mathbb{E}\|\sum_{i=1}^{n}\epsilon_i x_i x_i^T\|$$

$$= O\left(\sqrt{\frac{\sigma_k^2 n d \log(d)}{\epsilon^{\frac{2}{k}}}} + \frac{\sigma_k^2 d \log d}{\epsilon^{\frac{2}{k}}}\right),$$

where we use Theorem C.2, with $\nu = O(\sigma_k^2 n d \epsilon^{-\frac{2}{k}})$ and $L = O(\sigma_k^2 d \epsilon^{-\frac{2}{k}})$.

Note that $Q_k = O(\sigma_k \epsilon^{-\frac{1}{k}} + (1/\epsilon)\sqrt{d/n}$. As $R'$ is bounded by $Q_k^2$, we can apply Theorem C.1 to get that with probability at least $1 - \tau$, $R'/n$ is bounded as follows:

$$\frac{R'}{n} = O\left(\sqrt{\frac{\sigma_k^2 d \log d}{n\epsilon^{\frac{2}{k}}}} + \frac{\sigma_k^2 d \log d}{n\epsilon^{\frac{2}{k}}} + \sigma_4^2\sqrt{\frac{\log(\frac{1}{\tau})}{n}} + \frac{\sigma_k^2}{\epsilon^{\frac{2}{k}}}\frac{\log(\frac{1}{\tau})}{n} + \frac{1}{\epsilon^2}\frac{d}{n}\frac{\log(\frac{1}{\tau})}{n}\right)$$

$$= \frac{1}{\epsilon}O\left(\sqrt{\frac{d\log d}{n}}\sigma_k\epsilon^{1-\frac{1}{k}} + \frac{d\log d}{n}\sigma_k^2\epsilon^{1-\frac{2}{k}} + \sigma_4\epsilon\sigma_4\sqrt{\frac{\log(\frac{1}{\tau})}{n}} + \sigma_k^2\epsilon\epsilon^{1-\frac{2}{k}} + \frac{d}{n}\right)$$

$$\text{( Using } \frac{\log(\frac{1}{\tau})}{n} = O(\epsilon).\text{)}$$

$$= \frac{1}{\epsilon}O\left(\left(\left(\sqrt{\frac{d\log d}{n}} + \sigma_k\epsilon^{1-\frac{1}{k}} + \sigma_k\epsilon^{\frac{1}{2}-\frac{1}{k}}\sqrt{\frac{d\log d}{n}} + \sigma_4\epsilon + \sigma_4\sqrt{\frac{\log(\frac{1}{\tau})}{n}}\right)\right)^2\right)$$

$$= \frac{1}{\epsilon}O\left(\left(\left(\sqrt{\frac{d\log d}{n}} + \sigma_k\epsilon^{1-\frac{1}{k}} + \sigma_4\sqrt{\frac{\log(\frac{1}{\tau})}{n}}\right)\right)^2\right)$$

$$\text{( Using } \sigma_4\epsilon \leq \sigma_k\epsilon^{1-\frac{1}{k}} \text{ and } \sigma_k\epsilon^{\frac{1}{2}-\frac{1}{k}} = O(1).\text{)}$$

$$\leq \frac{\delta^2}{\epsilon},$$

where we use the parameter regime stated in Lemma F.3. $\qquad\square$

## F.4 Minimum Eigenvalue of Large Subsets

In this section, we prove that under bounded central moments, the minimum eigenvalue of $\Sigma_{S'}$, of each large enough subset $S'$, has a lower bound close to 1. Our result is similar in spirit to Koltchinskii and Mendelson [KM15, Theorem 1.3] that only bounds the eigenvalue of $\overline{\Sigma}_S$. The proof of the following lemma is very similar to the proof of Lemma F.3.

**Lemma F.5.** *Consider the setting in Lemma F.3. Then, for a constant $c > 0$, with probability $1 - \tau - \exp(-cn\epsilon)$, the following holds:*

$$\min_{S':|S'|\geq(1-\epsilon)n} v^T\overline{\Sigma}_{S'}v \geq 1 - \frac{\delta^2}{\epsilon},$$

*where $\delta = O(\sqrt{\frac{d\log d}{n}} + \sigma_k\epsilon^{1-\frac{1}{k}} + \sigma_4\sqrt{\frac{\log(\frac{1}{\tau})}{n}})$.*

*Proof.* Without loss of generality, assume that $\mu = 0$. We will assume that event $\mathcal{E}$ from Lemma 3.2 holds, with an additional probability of error $\exp(-cn\epsilon)$, that is

$$\sup_{v\in\mathcal{S}^{d-1}}\left|\{i : x_i^T v \geq Q_k\}\right| \leq n\epsilon.$$

Let $f$ be as defined in the proof of Lemma F.3. For a sequence $y_1, \ldots, y_n$, let $y_{(1)}, \ldots, y_{(n)}$ be its rearrangement in non-decreasing order. For any unit vector $v$, we have that

$$\min_{S': |S'| \geq (1-\epsilon)n} v^T \overline{\Sigma}_{S'} v \geq \min_{w \in \Delta_{n,\epsilon}} v^T \overline{\Sigma}_w v = \min_{w \in \Delta_{n,\epsilon}} \sum_{i=1}^n w_i (x_i^T v)^2$$

$$\geq \sum_{i=1}^{(1-\epsilon)n} (x_i^T v)_{(i)}^2 / ((1-\epsilon)n)$$

$$\geq \sum_{i=1}^n (f((x_i^T v)^2) - Q_k^2 \epsilon n) / ((1-\epsilon)n),$$

where we use that at most $\epsilon n$ points have projections larger than $Q_k^2$. Thus we get that the minimum eigenvalue of any large subset is lower bounded by:

$$\min_{w \in \Delta_{n,\epsilon}} \min_{v \in \mathcal{S}^{d-1}} \sum_{i=1}^n w_i (x_i^T v)^2 \geq \min_{v \in \mathcal{S}^{d-1}} \sum_{i=1}^n f((x_i^T v)^2) - Q_k^2 \epsilon n.$$

Let $h(\cdot)$ be the negative of the function $f(\cdot)$. Define the following random variable $Z$ and its counterpart $Z'$:

$$Z := \sup_{v \in \mathcal{S}^{d-1}} \sum_{i=1}^n h((x_i^T v)^2), \qquad Z' := \sup_{v \in \mathcal{S}^{d-1}} \sum_{i=1}^n h((x_i^T v)^2) - \mathbb{E} h((x_i^T v)^2)$$

From Lemma C.4, it follows that $|\mathbb{E} h((x_i^T v)^2) + 1| = |\mathbb{E} f((x_i^T v)^2) - 1| = O(\sigma_k^2 \epsilon^{1-\frac{2}{k}})$. This immediately gives us that

$$|Z' - Z - n| = O(n \sigma_k^2 \epsilon^{1-\frac{2}{k}}).$$

Therefore, the desired quantity satisfies the following inequalities:

$$(1-\epsilon)n \min_{w \in \Delta_{n,\epsilon}} \min_{v \in \mathcal{S}^{d-1}} \sum_{i=1}^n w_i (x_i^T v)^2 \geq \min_{v \in \mathcal{S}^{d-1}} \sum_{i=1}^n f((x_i^T v)^2) - Q_k^2 \epsilon n$$

$$= -\sup_{v \in \mathcal{S}^{d-1}} \sum_{i=1}^n h((x_i^T v)^2) - Q_k^2 \epsilon n$$

$$= -Z - Q_k^2 \epsilon n$$

$$\geq -Z' + n - O(n \sigma_k^2 \epsilon^{1-\frac{2}{k}}) - \epsilon Q_k^2 \epsilon n.$$

We thus require a high probability upper bound on $Z'$. Note that $Z'$ behaves similarly to $R'$, defined in the proof of Lemma F.3. Similar to the proof of Lemma F.4, we get that, with probability at least $1 - \tau$,

$$\frac{Z'}{n} \leq \frac{1}{\epsilon} \left( O\left( \sqrt{\frac{d \log d}{n}} + \sigma_k \epsilon^{1-\frac{1}{k}} + \sigma_4 \sqrt{\frac{\log(1/\tau)}{n}} \right) \right)^2.$$

Note that the remaining terms $\sigma_k^2 \epsilon^{1-\frac{2}{k}} = O(\delta^2/\epsilon)$ and $\epsilon Q_k^2 = O(\sigma_k^2 \epsilon^{-\frac{2}{k} + \frac{d}{n\epsilon}} = O(\delta^2/\epsilon)$. Therefore, we get the minimum eigenvalue of any large subset is at least

$$\min_{w \in \Delta_{n,\epsilon}} \lambda_{\min}(\overline{\Sigma}_w) \geq 1 - \frac{\delta^2}{\epsilon},$$

where $\delta = O(\sqrt{\frac{d \log d}{n}} + \sigma_k \epsilon^{1-\frac{1}{k}} + \sigma_4 \sqrt{\frac{\log(\frac{1}{\tau})}{n}})$.

$\square$

## F.5 Controlling the Mean

Lemmas F.3 and F.5 give a control on the second moment matrix. We will now further remove $O(\epsilon)$ fraction of points to obtain $w$ such that $\|\mu_w - \mu\|$ is small.

**Lemma F.6.** *Let $x_1, \ldots, x_n$ be $n$ i.i.d. random variables from a distribution with mean $\mu$ and covariance $\Sigma \preceq I$. Further, assume that the $x_i$'s are drawn from a distribution with $k$-th bounded central moment $\sigma_k$ for a $k \geq 4$. Let $u \in \Delta_{n,\epsilon}$. Assume that $\log(1/\tau)/n = O(\epsilon)$. Then, for a constant $c > 0$, the following holds with probability $1 - \tau - \exp(-cn\epsilon)$:*

$$\min_{w \in \Delta_{n,4\epsilon,u}} \| \sum_{i=1}^n w_i x_i - \mu \| = O(\sqrt{d/n} + \sigma_k \epsilon^{1-\frac{1}{k}} + \sqrt{\log(1/\tau)/n}).$$

*Proof.* Without loss of generality, let us assume that $\mu = 0$. Also, assume that the event $\mathcal{E}$ from Lemma 3.2 holds, with the additional error of $\exp(-cn\epsilon)$. Let $g(\cdot)$ be the following function:

$$g(x) = \begin{cases} x, & \text{if } x \in [-Q_k, Q_k] \\ Q_k, & \text{if } x > Q_k \\ -Q_k, & \text{if } x < -Q_k. \end{cases}$$

Let $N$ be the following random variable:

$$N = \sup_{v \in \mathcal{S}^{d-1}} \sum_{i=1}^n g(v^T x_i) = \sup_{v \in \mathcal{S}^{d-1}} \left| \sum_{i=1}^n g(v^T x_i) \right|,$$

where we use that $g(\cdot)$ is an odd function. We also define the following empirical process, where each term is centered:

$$N' = \sup_{v \in \mathcal{S}^{d-1}} \sum_{i=1}^n g(v^T x_i) - \mathbb{E}[g(v^T x_i)] = \left| \sup_{v \in \mathcal{S}^{d-1}} \sum_{i=1}^n g(v^T x_i) - \mathbb{E}[g(v^T x_i)] \right|.$$

As $Q_k = \Omega(\sigma_k \epsilon^{-1/k})$, Lemma C.4 states that $\sup_v \mathbb{E} g(v^T x) = O(\sigma_k \epsilon^{1-\frac{1}{k}})$, and this gives that

$$|N - N'| = O(n \sigma_k \epsilon^{1-\frac{1}{k}}).$$

We now use duality to write the following:

$$\min_{w \in \Delta_{n,\epsilon,u}} \| \sum_{i=1}^n w_i x_i \| = \min_{w \in \Delta_{n,\epsilon,u}} \max_{v \in \mathcal{S}^{d-1}} \langle \sum_{i=1}^n w_i x_i, v \rangle$$

$$= \max_{v \in \mathcal{S}^{d-1}} \min_{w \in \Delta_{n,\epsilon,u}} \langle \sum_{i=1}^n w_i x_i, v \rangle$$

$$\leq 5\epsilon Q_k + \left| \frac{1}{(1-\epsilon)n} N \right| \leq O(\epsilon Q_k) + O(\sigma_k \epsilon^{1-1/k}) + 2N',$$

where the last step uses Lemma D.4. We now use Lemma F.7 to conclude that $N'$ concentrates. Recall that $\epsilon Q_k = O(\sigma_k \epsilon^{1-\frac{1}{k}} + \sqrt{d/n})$. Overall, we get that, with probability $1 - \tau - \exp(-n\epsilon)$, there exits a $w \in \Delta_{n,\epsilon,u}$, such that $\| \sum w_i x_i \| = O(\sqrt{d/n} + \sigma_k \epsilon^{1-\frac{1}{k}} + \sqrt{\log(1/\tau)/n})$. $\qquad \square$

**Lemma F.7.** *Consider the setting in Lemma F.6. Then, with probability, $1 - \tau - \exp(-n\epsilon)$,*

$$\frac{R'}{n} = O\left( \sqrt{\frac{d}{n}} + \sqrt{\frac{\log(1/\tau)}{n}} + \sigma_k \epsilon^{1-\frac{1}{k}} \right).$$

*Proof.* We first calculate the wimpy variance required for Theorem C.1,

$$\sigma^2 = \sup_{v \in \mathcal{S}^{d-1}} \sum_{i=1}^n \mathbb{V}(g(x_i^T v)) \leq \sup_{v \in \mathcal{S}^{d-1}} \sum_{i=1}^n \mathbb{E} g(v^T x_i)^2 \leq \sup_{v \in \mathcal{S}^{d-1}} n \mathbb{E}(v^T x_i)^2 \leq n.$$

We use symmetrization, contraction of Rademacher averages to bound $\mathbb{E}R'$.

$$\mathbb{E}R' = \mathbb{E}\sup_{v\in\mathcal{S}^{d-1}}\sum_{i=1}^{n}g(v^Tx_i) - \mathbb{E}g(v^Tx_i)$$

$$\leq 2\mathbb{E}\left|\sup_{v\in\mathcal{S}^{d-1}}\sum_{i=1}^{n}\epsilon_i g(v^Tx_i)\right|$$

$$\leq 2\mathbb{E}\left|\sup_{v\in\mathcal{S}^{d-1}}\sum_{i=1}^{n}\epsilon_i v^Tx_i\right| = 2\mathbb{E}\|\sum_{i=1}^{n}\epsilon_i x_i\| \leq 2\sqrt{\frac{d}{n}}.$$

By applying Theorem C.1, we get that with probability at least $1 - \tau$,

$$\frac{R'}{n} = O\Big(\frac{\mathbb{E}R'}{n} + \sqrt{\frac{\log(1/\tau)}{n}} + Q_k\frac{\log(1/\tau)}{n}\Big)$$

$$= O\Big(\sqrt{\frac{d}{n}} + \sqrt{\frac{\log(1/\tau)}{n}} + \sigma_k\epsilon^{-\frac{1}{k}}\frac{\log(\frac{1}{\tau})}{n} + \frac{1}{\epsilon}\sqrt{\frac{d}{n}}\frac{\log(1/\tau)}{n}\Big)$$

$$= O\Big(\sqrt{\frac{d}{n}} + \sqrt{\frac{\log(1/\tau)}{n}} + \sigma_k\epsilon^{1-\frac{1}{k}}\Big),$$

where the last inequality uses the assumption that $\frac{\log(1/\tau)}{n} = O(\epsilon)$. $\qquad\square$

## F.6 Proof of the Main statement

We now combine the results in the previous lemmas to obtain the stability of a subset with high probability. Although we prove the following result showing the existence of $(2\epsilon', \delta)$ stable subset, this can generalized to existence of $(C\epsilon, O(\delta))$ stable subset for a large constant $C$.

**Theorem F.8.** *(Theorem 1.8) Let $S = \{x_1, \ldots, x_n\} \subset \mathbb{R}^d$ be $n$ i.i.d. points from a distribution with mean $\mu$ and covariance $\Sigma$ such that $(1 - 2\sigma_k^2\gamma^{1-\frac{1}{k}})I \preceq \Sigma \preceq I$. Further assume that for a $k \geq 4$, the $k^{th}$ central moment is bounded by $\sigma_k$. Let $\epsilon' = \Theta(\epsilon + \frac{\log(1/\tau)}{n}) \leq c$ for a sufficiently small constant $c$.*

*Then, with probability at least $1 - \tau$, there exists a subset $S' \subseteq S$ s.t. $|S'| \geq (1 - \epsilon')n$ and $|S'|$ is $(2\epsilon', \delta)$-stable with $\delta = O(\sigma_k\epsilon^{1-\frac{1}{k}} + \sqrt{\frac{d\log d}{n}} + \sigma_4\sqrt{\frac{\log(1/\tau)}{n}})$.*

*Proof.* First note that, for the bounded covariance condition, Theorem 1.4 already gives a guarantee that, with probability at least $1 - \tau$,

$$\|\widehat{\mu} - \mu\| = O\Big(\sqrt{(d\log d)/n} + \sqrt{\epsilon} + \sqrt{\log(1/\tau)/n}\Big). \tag{47}$$

Therefore, the guarantee of this theorem statement is tighter only in the following regimes:

$$\log(1/\tau)/n = O(\epsilon), \qquad O(\sigma_k\epsilon^{\frac{1}{2}-\frac{1}{k}}) = O(1), \qquad d\log d/n = O(\epsilon). \tag{48}$$

For the rest of the proof, we will assume that all three of these conditions hold. Similar to the proof of Theorem 1.4, we will first prove the statement when the samples are bounded. Without loss of generality, we will assume $\mu = 0$.

**Base case: Bounded support** In this case, we will assume that $\|x_i\| = O(\sigma_k\epsilon^{-1/k}\sqrt{d})$ almost surely. We will use Lemma F.2 to show that the set is stable. Set $\tilde{\epsilon} = \epsilon'/C'$ for a large enough constant $C'$ to be determined later.

Note that $x_1, \ldots, x_n$ satisfy the conditions of Lemmas F.5, F.3, and F.6. In particular, we will use Lemma F.5 with $C\tilde{\epsilon}$, where $C$ is large enough. By choosing $\epsilon' = \Omega(\log(1/\tau)/n)$, we get that, with probability $1 - \tau/3$, for any $S' : |S'| \geq (1 - C\tilde{\epsilon})n$ and unit vector $v$,

$$\frac{\sum_{i\in S'}(v^Tx_i)^2}{|S'|} \geq 1 - \frac{\delta^2}{C\tilde{\epsilon}}. \tag{49}$$

We first look at the variance using the guarantee in Lemma F.3: Let $u \in \Delta_{n,\tilde{\epsilon}}$ be the distribution achieving the minimum in Lemma F.3. By choosing $\epsilon' = \Omega(\log(1/\tau)/n)$, we get that with probability $1 - \tau/3$,

$$\sum_{i=1}^{n} u_i (x_i^T v)^2 \leq 1 + \frac{\delta^2}{\tilde{\epsilon}}. \tag{50}$$

We now obtain a guarantee on the mean using Lemma F.6. For this $u$, let $w \in \Delta_{n,4\tilde{\epsilon},u}$ be the distribution achieving the minimum in Lemma F.6. Then with probability $1 - \tau/3$,

$$\| \sum_{i=1}^{n} w_i x_i \| \leq \delta. \tag{51}$$

Since $u \in \Delta_{n,4\tilde{\epsilon},w}$ and $w \in \Delta_{n,\tilde{\epsilon},u}$, we have that $u \in \Delta_{n,5\tilde{\epsilon}}$. Moreover,

$$\sum_{i=1}^{n} w_i (x_i^T v)^2 \leq \sum_{i=1}^{n} \frac{u_i}{1-\tilde{\epsilon}} (x_i^T v) = \frac{1}{1-\tilde{\epsilon}} (1 + \frac{\delta^2}{\tilde{\epsilon}}) \leq 1 + \frac{1}{1-\tilde{\epsilon}} (\tilde{\epsilon} + \frac{\delta^2}{\tilde{\epsilon}}) \leq 1 + \frac{4\delta^2}{\tilde{\epsilon}}. \tag{52}$$

Therefore, we have that $u \in \Delta_{n,5\tilde{\epsilon}}$ and satisfies the requirements of Lemma F.2, where we note that $r_1 = O(1)$ and $r_2 = O(1)$ to get the desired statement. By a union bound, the failure probability is $\tau$. Finally, we choose $C$ and $C'$ large enough such that the cardinality of the stable set is at least $(1 - \epsilon')n$ and it is $(2\epsilon', \delta)$ stable.

**General case: Unbounded support** We first do a simple truncation. Let $E$ be the following event:

$$E = \{X : \|X\| \leq C\sigma_k \epsilon^{-\frac{1}{k}} \sqrt{d}\}. \tag{53}$$

Let $Q$ be the distribution of $X$ conditioned on $E$. Note that $P$ can be written as convex combination of two distributions: $Q$ and some distribution $R$,

$$P = (1 - \mathbb{P}(E))Q + \mathbb{P}(E^c)R. \tag{54}$$

Let $Z \sim Q$. Using Lemma C.5, we get that $\|\mathbb{E}Z\| \leq 2\sigma_k \epsilon^{1-\frac{1}{k}}/C^k$ and $(1 - 3\sigma_k^2 \epsilon^{1-\frac{2}{k}}/C^k) \preceq \text{Cov}(Z) \preceq I$. Thus the distribution $Q$ satisfies the assumptions of the base case for $C \geq 2$.

Let $S_E$ be the set $\{X_i : X_i \in E\}$. A Chernoff bound gives that given $n$ samples from $P$, with probability at least $1 - \exp(-n\epsilon')$,

$$E_1 = \{|S_E| \geq (1 - \epsilon'/2)n\}. \tag{55}$$

For a fixed $m \geq (1 - \epsilon'/2)n$, let $z_1, \ldots, z_m$ be $m$ i.i.d. draws from the distribution $Q$. Applying the theorem statement for $Q$, as it satisfies the base case above, we get that, with probability at least $1 - \exp(-cm\epsilon')$, $\exists S' \subset [m] : |S'| \geq (1 - \epsilon'/2)m \geq (1 - \epsilon'/2)^2 n \geq (1 - \epsilon')n$, such that $S'$ is $(2\epsilon', \delta')$-stable. This gives us a set $S'$ which is stable with respect to $\mathbb{E}Z$. Using triangle inequality, we get that the set $S'$ is $(\epsilon, \delta')$ stable with respect to $\mu$ as well, where $\delta' = \delta + \|\mu - \mathbb{E}Z\| = \delta + O(\sigma_k \epsilon^{1-\frac{1}{k}})$.

We can now marginalize over $m$ to get that with probability except $1 - 2\exp(-cn\epsilon')$, the desired claim holds. Choosing $\epsilon' = \Omega(\log(1/\tau)n)$, we can make probability of failure less than $\tau$. □

## Footnotes

[1]If $(x_1, \ldots, x_n)$ are i.i.d., then choosing the buckets $B_i = \{x_{(i-1)m}, \ldots, x_{im}\}$ for $i \in [k]$ preserves independence. In particular, any partition of $k$ sets of equal cardinality that does not depend on the values of $(x_1, \ldots, x_n)$ suffices. Therefore, Theorem B.1 and Theorem B.2 hold for this bucketing strategy too.