[Reviews · NeurIPS 2020]

Review 1

Summary and Contributions: The paper considers the problem of robust mean estimation when the data is both heavy-tailed and corrupted with outliers. In this adversarial setup, the authors show that existing estimators do get the near-optimal subgaussian non-asymptotic rate, with optimal dependence on the contamination level \epsilon under bounded 2nd moment condition. Moreover, when the covariance is known, then, their estimator gets the optimal \epsilon^{1-1/k} rate for k-the bounded central-moment. The key technical result is the show the existence of a "good/stable" subset which is at the heart of efficient robust estimators.

Strengths: I think the paper is technically strong, and the results are very significant. This shows that existing spectral-filtering based approaches do indeed give optimal rates for heavy-tailed estimation, which I think is very interesting. ### After Rebuttal #### I thank the authors for their thoughtful response. After looking at it and reading the other reviews, my score remains the same.

Weaknesses: Nothing major.

Correctness: Yes.

Clarity: Yes.

Relation to Prior Work: Yes.

Reproducibility: Yes

Additional Feedback: The authors state that [DL19,LLVZ19] work only in the additive contamination setting, and not in the strong contamination model. Why is that?


Review 2

Summary and Contributions: I thank the authors for their feedback. --- The paper presents a robust mean estimator with sub-Gaussian concentration.

Strengths: The estimator run in polynomial time.

Weaknesses: The authors did not explicitly specify the polynomial complexity, thus the algorithm may still be impractical.

Correctness: Yes

Clarity: Yes

Relation to Prior Work: Yes

Reproducibility: Yes

Additional Feedback:


Review 3

Summary and Contributions: CONTEXT The workhorse behind the robust statistics results from the last few years is an iterative filtering procedure that eventually identifies a subset of the corrupted data whose empirical mean is close to the true mean. This framework works under a "stability" assumption that the clean data satisfies the property that any large subset has empirical mean/covariance close to the true mean/covariance. Another line of work starting with Lugosi-Mendelson '19 has focused on getting sub-Gaussian rates for mean estimation of heavy-tailed (uncorrupted) distributions in high dimensions, e.g. for identity covariance distributions, the goal is to obtain \sqrt{d/n} + \sqrt{log(1/delta)/n} rates (as opposed to sqrt{d/(n\delta)} via empirical mean or \sqrt{d\log(1/\delta)/n} via geometric median). Subsequent works have culminated in estimators that achieve such optimal sub-Gaussian rates in near-linear time and that can also tolerate *additive* corruptions. ------------------------------------------------------------ THIS WORK This work is the first to get the best of both worlds by giving a refined analysis of the stability-based framework from the robust statistics literature, showing that (modulo a standard bucketing preprocessing step) the iterative filtering approach already achieves optimal sub-Gaussian rates for mean estimation, even in the presence of corruptions in the strong contamination model (where the adversary can both add and delete points). They also show that their analysis extends to the k-hypercontractive, identity-covariance case, at the loss of an extra \sqrt{log d} factor. ------------------------------------------------------------ TECHNIQUES The main technical contribution of this work is to strengthen the existing bound for the existence of a large stable subset in a collection of iid samples from a distribution with bounded covariance, in the high probability regime. The key idea is as follows. For w a set of high-min-entropy weights on the clean points and Sigma_w the weighted empirical covariance, a standard step in the robust mean estimation papers is to solve the minimax problem of searching for w minimizing ||Sigma_w - I|| (the max player picks a test matrix in the spectrahedron). Crucially, they pass to the maximin problem, which then becomes an empirical process question, where the empirical process is indexed over the spectrahedron, and then one can reason about concentration of min_w ||Sigma_w - I|| using Talagrand's concentration inequality. This shows the covariance bound in the definition of stability, and the mean deviation bound then follows by taking w minimizing ||Sigma_w - I|| and applying similar reasoning to refine w further. POST-AUTHOR FEEDBACK: Thanks for providing additional details on the sample complexity achieved by prior works. My opinion is unchanged, that is, I continue to be in favor for acceptance.

Strengths: This work is very nice for simultaneously 1) answering the open question of efficiently getting optimal sub-Gaussian rates for heavy-tailed mean estimation in the strong contamination model, and 2) showing that the existing filtering approach *already* achieves this (modulo a standard preprocessing step). The latter point ties together the two lines of work on robust mean estimation and heavy-tailed mean estimation in an especially satisfying way. While previous works like Cheng-Diakonikolas-Ge have looked at the dual SDP max_M min_w <Sigma_w,M>, they did so from an algorithmic perspective by using it as part of their descent procedure, whereas the nice idea of this work is to bound the value of this SDP in the *analysis*, for the sake of showing the existence of a stable set.

Weaknesses: The only "limitations" are that 1) they still need to bucket before running iterative filtering on the bucketed averages for eps = o(1), and 2) the results for k-hypercontractive are not yet optimal, but these aren't really weaknesses so much as interesting follow-up questions.

Correctness: I have verified that the proofs are correct.

Clarity: The paper is very cleanly written, and the steps of the argument are quite easy to follow, especially because they do a good job providing intuitive explanations for the main steps.

Relation to Prior Work: The author(s) clearly explain the relation of this work to previous works.

Reproducibility: Yes

Additional Feedback: It would be good to flesh out lines 140 to 143 a bit more. In particular, it would be good to directly compare the bound of Theorem 1.4 to the deterministic regularity condition bounds from the resilience papers, DHL19, DKK+17/DK19, etc. Minor typos in supplement: - equation in 310: N/((1-eps)n) -> 2N/((1-eps)n) - equation in line 631 should use Q_0 - equation in line 686: w^1 -> w


Review 4

Summary and Contributions: This paper addresses the problem of computationally efficient robust mean estimation from a (potentially adversarially) corrupted sample of n datapoints with the properties of: 1. Optimal dimension dependence of order \sqrt{d/n} 2. Optimal additive order \sqrt{epsilon} dependence where epsilon is the fraction of corrupted points; and 3. Optimal subgaussian rates of sqrt{log(1/tau)/n} where tau is the failure probability. In this context the paper makes the following contributions. 1. They show that with exponentially high probabilty, a sufficiently large subset of i.i.d. samples satisfies stability (defined in DK) with optimal epsilon rate and subgaussian dependence, and near-optimal dimension dependence (d log d rather than d). This implies (see DK) the existence of a polynomial time algorithm to obtain a near-optimal estimate of mu (in the sense of 1, 2, 3 above) 2. This is upgraded to the optimal rate (in terms of dimension) using a simple preprocessing step of running the algorithm, not on the original data, but on the empirical of a random bucketing of the data. This analysis relies also on stability, and the bucketing idea is from the median of means work. This provides the first computationally efficient procedure to satisfy all of 1, 2, 3, above. 3. However, the stability analysis earlier is not 'instance optimal' in the following way. When the distribution has additional favorable properties like higher order moments that improve the estimation rate in terms of epsilon. The paper establishes a stronger stability condition in this setting, again yielding a poly time estimator. Reference key: DK: Diakonikolas and Kane "recent advances ...." DKK+: Diakonikolas et al 'robust estimators in high ...'

Strengths: - Robust mean estimation from contaminated data is a problem of central interest to the machine learning and statistics community. This paper will be of interest to the NeurIPS audience. - The main strength of the paper is focusing on the stability structural property, as a method to obtain a robust estimator and computationally efficient estimator. - The paper also reduces stability via Claim 2.1 to control of the mean and (generalized) second moment. These conditions are then established using empirical process tools like Talagrand's concentration. This may be of independent interest.

Weaknesses: I do not have significant points of weaknesses for the paper. I have some comments on presentation and clarity in other portions of the review.

Correctness: At this point I do not see significant correctness issues with the paper.

Clarity: The paper is generally well-written and relatively easy to follow. Some comments: 1. There is inconsistent use of big O notation, and constants C, c notation. I would recommend to use as far as possible, one of them and, if possible, the latter. Some examples are in the additional comments section. 2. Def 1.2: Why is the generalized covariance compared to identity? Should this not be defined strictly for covariance = identity? Otherwise, if Sigma is rank deficient, stability would not hold for non trivial delta.

Relation to Prior Work: Related work is appropriately cited and introduced throughout the paper.

Reproducibility: Yes

Additional Feedback: Post author feedback, I realized I did not put these comments in the review. Apologies! - L.19 'highly suboptimal' is odd phrasing. This is repeated in a number of places. - L.140 why would epsilon' not be order one, it is anyway bounded by 1. Perhaps this is to be \Theta(1)? - L. 157 'non constructive' should probably be 'computationally inefficient' - Something is off in equation 4, there is w on the LHS but not the right where it is an optimization variable in the min max formula. Probably this is the second moment discrepancy from claim 2.1? EDIT POST RESPONSE: After reading the author response, I am maintaining my score on the paper.

[Author Response · NeurIPS 2020]

We thank the reviewers for their time and effort in providing feedback. We are encouraged by the universally positive
scores, and that all the reviewers appreciated the paper for the following: (i) significant results (**R1**,**R3**,**R4**), (ii) technical
contribution (**R1**,**R3**,**R4**), (iii) a unified view of heavy-tailed and robust mean estimation (**R3**), and (iv) clarity (**R1**, **R2**,
**R3**). For completeness, we summarize the contributions of our paper below.

**Summary:** Our goal is to show that a host of recent *computationally efficient* algorithms achieve optimal (or near-
optimal) statistical results for two important families of distributions. We achieve this by showing that the underlying
deterministic structural condition, *stability*, holds with optimal (or near-optimal) rates. Thus, our work simultaneously
improves the statistical error guarantee of these existing algorithms without designing a new algorithm.

We address the individual questions and comments by the reviewers below.

**Reviewer 1 (R1):** We thank the reviewer for the positive feedback. Regarding the tolerance of [DL19,LLVZ19] to
adversarial contamination: The algorithms and analyses in these papers establish tolerance to additive contamination
but not strong contamination. More broadly, there is a significant difference between additive and strong contamination.

**Reviewer 2 (R2):** We thank the reviewer for the positive feedback. The reviewer asked regarding the *polynomial*
*complexity* and *practicality* of known stability-based algorithms. We would like to emphasize that the computational
aspects of these algorithms ([DK19, CDG18, SCV18, DKK+17,DHL19, CDG20]) are well-studied. For concreteness,
we specify the running time for two existing algorithms that achieve the rate in Proposition 1.6:

- Universal filter [DK19]: $\tilde{O}(\min(k,d)k^2 d)$
- Quantum entropy filter [DHL19]: $\tilde{O}(k^2 d)$

As shown in [DKK+17, DHL19], the filter algorithm (and its variants) is **scalable** and **practical**. In particular, these
algorithms have been successfully implemented in practical applications (see [DHL19] and [DKK+17] for experiments).
Combining our statistical results with the runtime of these filtering algorithms, we obtain fast algorithms for heavy-tailed
robust mean estimation in the strong contamination model. Prior to our work, no polynomial-time algorithm (with
provable guarantees) was known in the strong contamination model.

**Reviewer 3 (R3):** We thank the reviewer for a detailed and encouraging feedback. We agree that it is an important
question, both conceptually and practically, if the stability-based algorithms achieve the optimal rate without pre-
processing.

**Comparison with prior work**: Additional details for lines $140-143$: Some prior works state their guarantees in terms
of sample complexity to get $O(\sqrt{\epsilon})$ error, either with constant probability or with probability $1-\tau$. In this terminology,
our sample complexity is $n = \Omega((d\log d + \log(1/\tau))/\epsilon)$. We mention the rates from the prior work below, all of which
are sub-optimal in one or more parameters:

- [DKK+17]: Guarantees are stated for large constant probability.
- Goodness [DHL19]: $\delta = \sqrt{(d\log d)/(n\tau)} + \sqrt{\epsilon}$.
- Resilience [SCV18]: Even for constant probability, the sample complexity is $\Omega(d^{3/2}/\epsilon + d/\epsilon^2)$.
- Generalized resilience [ZJS19]: They give two bounds: (i) $\delta = O(\sqrt{\epsilon} + \sqrt{\frac{d\log(d/\tau)}{n}})$, (ii) $\delta = \sqrt{\frac{\epsilon}{\tau}} + \frac{1}{\tau}\sqrt{d/n}$.
- [PBR19]: $\delta = O(\sqrt{d\log(d/\tau)/n})$.

Thanks for pointing out the typos, we will fix them.

**Reviewer 4 (R4):** We thank the reviewer for the detailed feedback.

**Presentation**: We would work on the provided suggestions. "*Some examples are in the additional comments section*" —
it seems, unfortunately, that this field is missing from the review. We will be happy to address these points once the
reviewer updates their review with these comments after the rebuttal phase.

**Rank-deficient $\Sigma$:** Why is the generalized covariance compared to identity?

We note that for rank-deficient $\Sigma$, the tightest bound that we can show is $\delta = O(\sqrt{\epsilon})$ (Theorem 1.4). This is precisely
because for this choice of $\delta$, the eigenvalue has trivial lower bound of $1 - \delta^2/\epsilon$, which is negative (see proof of Claim
2.1). As the reviewer points out, we need more information about $\Sigma$ to obtain $o(\sqrt{\epsilon})$ error. For such cases, we assume
the knowledge of $\Sigma$, and obtain tighter rates in Theorem 1.8.

[Meta-Review · NeurIPS 2020]

The reviewers were unanimously enthusiastic about your paper. The reviewers have made several expository suggestions that we hope you will incorporate in the final version of your paper.